# A Text GAN for Language Generation with Non-Autoregressive Generator

## Abstract

Despite the great success of Generative Adversarial Networks (GANs) in generating high-quality images, GANs for text generation still face two major challenges: first, most text GANs are unstable in training mainly due to ineffective optimization of the generator, and they heavily rely on maximum likelihood pretraining; second, most text GANs adopt autoregressive generators without latent variables, which largely limits the ability to learn latent representations for natural language text. In this paper, we propose a novel text GAN, named NAGAN, which incorporates a non-autoregressive generator with latent variables. The non-autoregressive generator can be effectively trained with gradient-based methods and free of pretraining. The latent variables facilitate representation learning for text generation applications. Experiments show that our model is competitive compared with existing text GANs in unconditional text generation, and it outperforms existing methods on sentence manipulation in latent space and unsupervised text decipherment.

## 1 Introduction

Generative Adversarial Networks (GANs) (Goodfellow et al., 2014) have achieved great success in generating continuous data, such as images with high resolution and fidelity (Brock et al., 2019). Unsurprisingly, GANs are also widely studied for text generation, but the adaptation is by no means trivial. The mainstream text GANs (Yu et al., 2017b; Guo et al., 2018) apply a different framework tailored for discrete sequence data, but there are remaining several unsolved research problems.

One problem lies in ineffective optimization. Most text GANs resort to gradient-free RL (reinforcement learning) algorithms, mainly due to the nature of discrete text data. However, since RL methods abandon the gradient information, they suffer from unstable training processes (Ke et al., 2019). Though some works (Chen et al., 2018) explored the feasibility of gradient-based methods, the optimization is still ineffective. As a result, most text GANs heavily rely on MLE pretraining, and some even report worse performance after GAN training (Caccia et al., 2020).

Another problem can be attributed to the generative model. Most text GANs adopt an autoregressive generator, which defines an explicit likelihood without any latent variable. Latent variables have empowered image GANs with various applications, such as unsupervised style transfer (Taigman et al., 2017) and image editing (Brock et al., 2017). However, most text GANs merely generate sentences from the learned distribution with autoregressive decoding, thereby hardly applicable to text style transfer or controlled text generation which may require latent representations.

We therefore challenge the conventional design of existing text GANs and argue that incorporating a non-autoregressive generator can benefit from both efficient gradient-based optimization methods and the use of latent variables. Our proposed model, named Non-Autoregressive GAN (NAGAN), consists of a non-autoregressive generator and a regularized discriminator. The non-autoregressive generator naturally translates latent variables to the tokens in parallel, and the gradient-based optimization on our feed-forward structure is significantly more effective than the same method on an autoregressive generator. The discriminator is regularized by Max Gradient Penalty (Zhou et al., 2019), which is another key to effective optimization. Our contributions are summarized as follows:

- We propose non-autoregressive GAN, which characterizes itself by employing a non-autoregressive generator and latent variables, and efficiently training from scratch with gradient-based methods. To our knowledge, NAGAN is the first text GAN which learns latent representations from scratch.

Table 1: Differences between various GANs. **Autoregressive**: using autoregressive generators. **Explicit**: using explicit generative models. **Latent**: equipped with latent variables. **Pretrain**: pretraining required or not. The mainstream TextGANs include Yu et al. (2017b); Guo et al. (2018); Shi et al. (2018); Che et al. (2017); Lin et al. (2017); Fedus et al. (2018).

| Model | Data | Autoregressive | Explicit | Latent | Optimization | Pretrain |
|---|---|---|---|---|---|---|
| Vanilla GAN (Goodfellow et al., 2014) | Image | ✗ | ✗ | ✓ | Gradient-based | ✗ |
| SOTA ImageGANs (Brock et al., 2019) | Image | ✗ | ✗ | ✓ | Gradient-based | ✗ |
| Mainstream TextGANs [*] | Text | ✓ | ✓ | ✗ | RL | ✓ |
| ScratchGAN (de Masson d'Autume et al., 2019) | Text | ✓ | ✓ | ✗ | RL | ✗ |
| RelGAN (Nie et al., 2019) | Text | ✓ | ✓ | ✗ | Gradient-based | ✓ |
| FMGAN (Chen et al., 2018) | Text | ✓ | ○ [1] | ✓ | Gradient-based | ✓ |
| NAGAN(Ours) | Text | ✗ | ✗ | ✓ | Gradient-based | ✗ |

- We conduct experiments on synthetic and real data, and show that NAGAN without MLE pretraining is competitive in unconditional text generation compared with the pretrained text GANs.
- By taking advantage of the latent variables and the non-autoregressive generator, our model can be applied to sentence manipulation in latent space and unsupervised text decipherment, where NAGAN significantly outperforms previous methods.

## 2 MOTIVATION OF NON-AUTOREGRESIVE GENERATION IN TEXT GANS

Non-autoregressive (NAR) generators have been widely used in image GANs (Goodfellow et al., 2014) and achieve great success in generating high-quality images (Brock et al., 2019). However, for generating texts, text GANs apply a different framework where autoregressive (AR) generators are used. In this section, we will discuss the differences between image and text GANs, and show why we need non-autoregressive generators in text generation.

### 2.1 GENERATIVE MODELS

Image and text GANs differ substantially in generative models. The image GAN is an `implicit` generative model, where a sample $\mathbf{x}$ is generated in two steps:

$$\mathbf{z} \sim p(\mathbf{z}); \qquad \mathbf{x} = G(\mathbf{z}) \tag{1}$$

where $\mathbf{z}$ is a latent variable, and $p(\mathbf{z})$ is the prior distribution. $G$ is a deterministic function from the latent space to the data space, usually parameterized by a NAR generator, where each pixel of $\mathbf{x}$ is generated simultaneously.

A text GAN is usually an `explicit` generative model. A discrete sequence sample $\boldsymbol{x} = [\boldsymbol{x}_1, \boldsymbol{x}_2, \cdots, \boldsymbol{x}_L]$ is sampled by a stochastic process from the distribution $P_{\mathcal{G}}(\boldsymbol{x})$, where

$$P_{\mathcal{G}}(\boldsymbol{x}) = \prod_{i=1}^{L} P_{\mathcal{G}}(\boldsymbol{x}_i | \boldsymbol{x}_1, \cdots, \boldsymbol{x}_{i-1}). \tag{2}$$

Eq (2) shows $\mathcal{G}$ is an autoregressive model, where tokens are sampled sequentially conditioned on previously generated prefixes. Most text GANs do not adopt latent variables, partially because Eq (2) is convenient for the MLE pretraining. There are also some explorations (Chen et al., 2018) on equipping text GANs with latent variables, but elaborate VAE-like pretraining is required.

However, directly injecting latent variables into the explicit generative model of text GANs can be problematic. For instance, when defining $P_{\mathcal{G}}(\boldsymbol{x}) = \mathbb{E}_{\boldsymbol{z} \sim p(\boldsymbol{z})} \prod_{i=1}^{L} P_{\mathcal{G}}(\boldsymbol{x}_i | \boldsymbol{x}_{<i}, \boldsymbol{z})$, we may face two problems: (1) Solely optimizing $P_{\mathcal{G}}$ does not make the model learn the latent representations. Without further constraints[2], the model can degenerate and simply ignores $z$, which becomes almost a vanilla language model, even if the generation distribution $P_{\mathcal{G}}$ perfectly fits the real distribution. (2) The representation in AR text GANs can be hardly applied to downstream tasks. In image GANs, $\mathbf{x}$ is fully determined by the sampled $\mathbf{z}$, so we can control the generated images by manipulating the latent variable. Moreover, the mapping from the latent space to the data space is continuous and differentiable, thereby facilitating applications such as image editing (Brock et al., 2017). In text

---

[1]FMGAN uses explicit generative models in pretraining and implicit ones in GAN training.

[2]For example, the reconstruction loss in VAEs alleviates degeneration. More discussed in Appendix A.1.1.

GANs, translating $z$ to $x$ is still a stochastic process when the latent variable is fixed, indicating that the generated sentences may not be entirely controllable. The non-differentiable generator can also be an obstacle for downstream applications.

Based on the above reasons, we explore the feasibility of adopting an implicit generative model with a NAR generator for text GANs. The NAR generator can be applied to various applications conveniently and will not degenerate into a language model (more discussed in Appendix A.1.1).

## 2.2 OPTIMIZATIONS

In image GANs, the generator $G_\theta$ and the discriminator $D_\phi$ play a minimax game:

$$\min_\theta \max_\phi V(D_\phi, G_\theta) = \mathbb{E}_{\mathbf{x} \sim p_{data}(\mathbf{x})}[\log D_\phi(\mathbf{x})] + \mathbb{E}_{\mathbf{z} \sim p(\mathbf{z})}[\log(1 - D_\phi(G_\theta(\mathbf{z})))]. \tag{3}$$

Since $G_\theta$ and $D_\phi$ are fully differentiable, they can be optimized alternately by gradient-based methods. In the text GANs, there is usually no latent variable, and the minimax game becomes

$$\min_\theta \max_\phi V(\mathcal{D}_\phi, \mathcal{G}_\theta) = \mathbb{E}_{\boldsymbol{x} \sim P_{data}(\boldsymbol{x})}[\log \mathcal{D}_\phi(\boldsymbol{x})] + \mathbb{E}_{\boldsymbol{x} \sim P_{\mathcal{G}_\theta}(\boldsymbol{x})}[\log(1 - \mathcal{D}_\phi(\boldsymbol{x}))]. \tag{4}$$

The discriminator $\mathcal{D}_\phi$ can be trained by gradient-based methods, but the output of $\mathcal{G}_\theta$ is discrete, so the gradient cannot be passed from $\mathcal{D}_\phi$ back to $\mathcal{G}_\theta$. Instead, the generator $\mathcal{G}_\theta$ can be optimized by the gradient-free REINFORCE algorithm (Williams, 1992):

$$\nabla_\theta \mathbb{E}_{\boldsymbol{x} \sim P_{\mathcal{G}_\theta}(\boldsymbol{x})}[R(\boldsymbol{x})] = \mathbb{E}_{\boldsymbol{x} \sim P_{\mathcal{G}_\theta}(\boldsymbol{x})}[R(\boldsymbol{x}) \nabla_\theta \log P_{\mathcal{G}_\theta}(\boldsymbol{x})], \tag{5}$$

where $R(\boldsymbol{x})$ is a reward function measuring the fitness of a generated sequence. If $R(\boldsymbol{x}) = \log(1 - \mathcal{D}(\boldsymbol{x}))$, it recovers the second term of Eq (4). Existing text GANs also devise various rewards tailored for text data (Guo et al., 2018; Shi et al., 2018; Lin et al., 2017; Fedus et al., 2018; Xu et al., 2018).

Gradient-free RL algorithms naturally suffer from high variance. In text GANs, the huge action spaces and the changing reward function[3] exacerbate the problem, so most text GANs heavily rely on the MLE pretraining. Some text GANs improve the training algorithms from the perspective of RL (Guo et al., 2018; Shi et al., 2018), but recent empirical studies show the RL training does not always improve the performance over the pretraining (Caccia et al., 2020; Semeniuta1 et al., 2018). ScratchGAN (de Masson d'Autume et al., 2019) devises improved RL techniques, for the first time freeing text GANs from MLE pretraining by increasing the batch size (~10x) and computation cost.

Some works (Kusner & Hernández-Lobato, 2016; Chen et al., 2018) explore the gradient-based methods and adopt continuous relaxations or gradient estimators for discrete text data. However, these models still rely on pretraining since the optimization is ineffective as well.

Different from existing works that focus on improvements of RL or the approximation methods, we find that one reason for the ineffective optimization may come from the generator architecture. To obtain gradients, an autoregressive generator with the gradient-based optimization method has to apply the gradient estimator recurrently, because the generator reads discrete tokens as inputs. In other words, the gradient flow from the last token will be approximated multiple times when it reaches the start of the sequence. On the contrary, a non-autoregressive generator has a feed-forward structure, and only one approximation is required at the last layer of the generator.

## 2.3 NON-AUTOREGRESSIVE TEXT GENERATION

Non-autoregressive text generation is first proposed for machine translation to generate the whole sequence in parallel with low latency (Gu et al., 2018; Ma et al., 2019). Most non-autoregressive generators are trained with cross-entropy loss, which assumes the tokens are conditional independent of each other when the input is given. As a result, the non-autoregressive generators can only be applied to the tasks where the outputs can be fully determined by the inputs (e.g., machine translation), and they are failed to capture complex distributions, known as the *multimodality problem* (Gu et al., 2018). Most non-autoregressive models require knowledge distillation to reduce the dataset complexity, otherwise they will experience a serious performance drop (Zhou et al., 2020). We assume that a key problem behind the phenomenon is the unreasonable independence assumption, so we replace

---

[3]The reward is estimated by the discriminator, and it keeps changing during training even for the same $\boldsymbol{x}$.

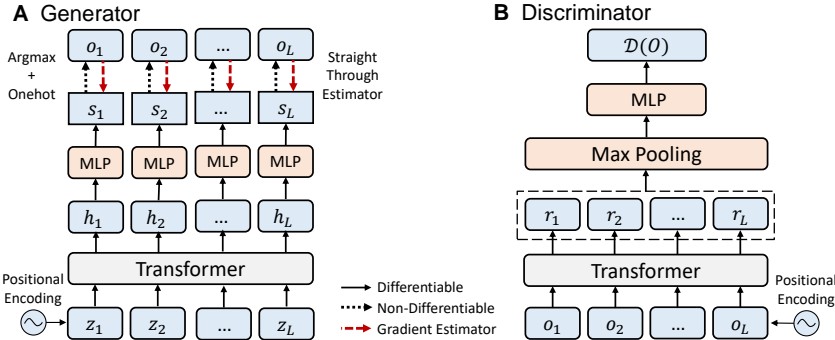

Figure 1: Architecture of NAGAN. (A) The generator converts the latent variable $\boldsymbol{Z} = [\boldsymbol{z}_1\boldsymbol{z}_2\ldots\boldsymbol{z}_L]$ to a sentence $\boldsymbol{O} = [\boldsymbol{o}_1\boldsymbol{o}_2\ldots\boldsymbol{o}_L]$, where each $\boldsymbol{o}_i$ is a one-hot vector. The gradient of non-differential operation is estimated by the straight through estimator. (B) The discriminator produces a score $D(\boldsymbol{O})$ for the sentence $\boldsymbol{O}$. The gradient from the discriminator can be passed back to the generator.

the cross-entropy loss with the GANs objective. The GAN objective considers the whole sequence and punishes implausible sentences, which does not rely on the independence assumption. In this paper, we explore the feasibility of generating diverse sentences with a non-autoregressive generator.

## 3 NON-AUTOREGRESSIVE GAN

As shown in Figure 1, our proposed non-autoregressive GAN (NAGAN) consists of a non-autoregressive generator with latent variables and a regularized discriminator. The framework is very similar to image GANs but differs from the mainstream text GANs substantially in two aspects: (1) We use implicit non-autoregressive generative models equipped with latent variables. (2) We use the gradient-based optimization method with Max Gradient Penalty to stabilize the training process.

### 3.1 GENERATOR

Our implicit generative model is defined by a sampling process. **First**, we sample the latent variable from the prior distribution. Instead of a single latent vector, we sample a sequence of continuous latent variables $\boldsymbol{Z} = [\boldsymbol{z}_1\boldsymbol{z}_2\cdots\boldsymbol{z}_L]$, where $L$ is a desired length of the target sequence[4] and each $\boldsymbol{z}_t$ is sampled from $\mathcal{N}(\boldsymbol{0}, \boldsymbol{I})$ independently for $1 \leq t \leq L$. This idea is inspired by Ma et al. (2019), which provides a good way for the non-autoregressive generator to leverage the latent variables. **Then**, the non-autoregressive generator $\mathcal{G}$ converts $Z$ to a sequence $\boldsymbol{O} = [\boldsymbol{o}_1\boldsymbol{o}_2\cdots\boldsymbol{o}_L]$, which is the one-hot representation of the generated sentence. We use one-hot representations here for the convenience of obtaining gradients. The process can be formulated as follows:

$$\boldsymbol{Z} \sim P(\boldsymbol{Z}); \qquad \boldsymbol{O} = \mathcal{G}(\boldsymbol{Z}). \tag{6}$$

$\mathcal{G}$ is implemented by a Transformer network (Gu et al., 2018; Vaswani et al., 2017):

$$[\boldsymbol{h}_1\boldsymbol{h}_2\cdots\boldsymbol{h}_L] = \text{Transformer}([\boldsymbol{z}_1\boldsymbol{z}_2\cdots\boldsymbol{z}_L]), \tag{7}$$

$$\boldsymbol{s}_t = \text{MLP}(\boldsymbol{h}_t) \in \mathbb{R}^V, \qquad \boldsymbol{o}_t = \text{onehot}(\arg\max_v(\boldsymbol{s}_{t,v})), \tag{8}$$

where $V$ is the size of the vocabulary, $\boldsymbol{s}_{t,v}$ is the $v$-th dimension of $\boldsymbol{s}_t$, and $\boldsymbol{o}_t \in \{0, 1\}^V$.

**Straight-Through Estimator.** The argmax operation is non-differentiable such that we cannot pass gradient from $\boldsymbol{o}_t$ back to the parameters of $\mathcal{G}$. Previous work (Kusner & Hernández-Lobato, 2016) has introduced the straight-through estimator (Bengio et al., 2013) to solve the problem. Given a scalar temperature $\tau$, if $\tau \to 0$, we have $\frac{\partial \boldsymbol{o}_t}{\partial \boldsymbol{s}_t} \approx \frac{\partial \text{softmax}(\boldsymbol{s}_t/\tau)}{\partial \boldsymbol{s}_t}$. Note that the straight-through estimator is only used for obtaining gradients, and the forward pass remains unchanged as Eq (8).

---

[4]$L$ can be different for each sequence. See more details in Section 3.3 and Appendix A.1.2.

## 3.2 DISCRIMINATOR

Similar to the vanilla GANs, the discriminator is trying to distinguish the real sentences and the generated sentences as a binary classification. Once receiving the generated sentence $\boldsymbol{O} = [\boldsymbol{o}_1 \boldsymbol{o}_2 \cdots \boldsymbol{o}_L]$, the discriminator produces $\mathcal{D}(\boldsymbol{O})$ as a score for the sentence. We simply choose Transformer for our discriminator. Some other architectures are discussed in Section 4.2.

$$[\boldsymbol{r}_1 \boldsymbol{r}_2 \cdots \boldsymbol{r}_L] = \text{Transformer}([\boldsymbol{o}_1 \boldsymbol{o}_2 \cdots \boldsymbol{o}_L]), \tag{9}$$

$$\mathcal{D}(\boldsymbol{O}) = \text{MLP}(\text{Max\_Pooling}([\boldsymbol{r}_1 \boldsymbol{r}_2 \cdots \boldsymbol{r}_L])). \tag{10}$$

**Max Gradient Penalty**. The gradient vanishing (Arjovsky et al., 2017) problem is a critical factor for causing the unstable training process in image GANs. When an unregularized discriminator is perfectly trained, the gradient becomes zero and cannot provide signals for the generator's optimization. We borrow a common training technique named Max Gradient Penalty from image GANs (Zhou et al., 2019). The technique restricts the Lipschitz constant of the discriminator, which is proven effective in gradient-based GAN's optimization. Max Gradient Penalty can be formulated as follows:

$$L_{GP} = \max_{\boldsymbol{O}'} \left\| \frac{\partial \mathcal{D}(\boldsymbol{O}')}{\partial \boldsymbol{O}'} \right\|^2, \tag{11}$$

where $\boldsymbol{O}'$ are sampled uniformly along the straight lines between pairs of points sampled from the real data and the generated data (Gulrajani et al., 2017).

## 3.3 TRAINING

The discriminator and the generator will be optimized alternatively, where the discriminator tries to distinguish the real and generated sentences, and the generator is updated to achieve a higher score $\mathcal{D}(\mathcal{G}(\boldsymbol{Z}))$. We utilize the GAN objective with non-saturating loss (Goodfellow et al., 2014).

$$\mathcal{L}_{\mathcal{D}} = -\mathbb{E}_{\boldsymbol{x} \sim P_{\text{data}}} \log \mathcal{D}(\text{onehot}(\boldsymbol{x})) - \mathbb{E}_{\boldsymbol{Z} \sim P(\boldsymbol{Z})} \log(1 - \mathcal{D}(\mathcal{G}(\boldsymbol{Z}))), \tag{12}$$

$$\mathcal{L}_{\mathcal{G}} = -\mathbb{E}_{\boldsymbol{Z} \sim P(\boldsymbol{Z})} \log \mathcal{D}(\mathcal{G}(\boldsymbol{Z})). \tag{13}$$

We further apply Max Gradient Penalty on the objective, and the loss of the discriminator becomes $\mathcal{L}'_{\mathcal{D}} = \mathcal{L}_{\mathcal{D}} + \lambda \mathcal{L}_{GP}$, where $\lambda$ is a hyper-parameter balancing the regularization terms. This objective can be regarded as a special case of Lipschitz GANs (Zhou et al., 2019).

With the straight-through estimator, both the generator and the discriminator are differentiable, so $\mathcal{L}'_{\mathcal{D}}$ and $\mathcal{L}_{\mathcal{G}}$ can be optimized alternately by gradient-based methods. The training steps are consistent with the vanilla GANs but essentially different from previous text GANs: we do not use MLE pretraining or sampling techniques devised for RL.

**Predicting the Length**. Non-autoregressive generators require a sentence length $L$ before generation in parallel. Existing non-autoregressive generators (Gu et al., 2018) usually leverage a classifier $P(L|C)$ to predict $L$ conditioned on the input $C$ (e.g., the source sentence in machine translation). In unconditional text generation, we directly estimate $P(L)$ by counting the lengths of the training samples since $C$ is not provided. During training and inference, $L$ is sampled from the same $P(L)$.

## 3.4 APPLICATIONS

**Manipulate Sentences in Latent Space.** To show the utility of latent variables, we introduce a sentence editing task: given a sentence (whose latent representation is $\boldsymbol{Z}_s$) which contains a source word, the task aims to obtain a new latent representation $\boldsymbol{Z}_t$, which can be translated to a sentence that contains a target word. Some cases are shown in Table 6. A previous method (Zhao et al., 2018) modifies the latent variable with an offset vector, which is obtained by subtracting the mean latent variable of the source word from that of the target word. This method ignores the context around the edited word and thus leads to a low success rate. More details are described in Appendix A.5.

NAGAN can use the gradient information to make specific modification for different contexts. If we want to modify the $t$-th word $\boldsymbol{x}_t$ in the source sentence to a desired word $\boldsymbol{x}'$, we can apply gradient descent on the latent variable to maximize the $\boldsymbol{x}'$-th dimension of $\boldsymbol{o}_t$ in Eq (8). One descent step can

be formulated as $\boldsymbol{Z}_s := \boldsymbol{Z}_s + \beta \frac{\partial \boldsymbol{o}_{t,\boldsymbol{x}'}}{\partial \boldsymbol{Z}_s}$, where $\beta$ is a small weight. The process will be repeated until success or exceeding a pre-specified iteration number. Note that the two methods are only applicable to the models with latent variables, where most text GANs are not applicable in this task.

**Unsupervised Decipherment.** Our model can apply to unsupervised decipherment (Yang et al., 2018) within the framework of CycleGAN (Zhu et al., 2017). The task provides two unparallel corpora: the plaintext $\mathcal{X}$ and the ciphertext $\mathcal{Y}$, where we aim to learn a mapping $F_{\mathcal{Y} \to \mathcal{X}}$ to decrypt the ciphertext. In order to adapt NAGAN to this task, we introduce an encoder $E$ to encode the data to a shared latent space and define $F_{\mathcal{Y} \to \mathcal{X}} := \mathcal{G}(E(\boldsymbol{y}, c_{\mathcal{Y}}), c_{\mathcal{X}})$, where $\boldsymbol{y} \in \mathcal{Y}$, $c_{\mathcal{Y}}$ and $c_{\mathcal{X}}$ are labels indicating the type for the source and target text. In a similar way, $F_{\mathcal{X} \to \mathcal{Y}} := \mathcal{G}(E(\boldsymbol{x}, c_{\mathcal{X}}), c_{\mathcal{Y}})$. Following Dai et al. (2019) , we utilize three losses: the cycle loss, the adversarial loss, and the reconstruction loss. We first define the three losses on $\mathcal{X}$, and the losses on $\mathcal{Y}$ can be obtained similarly. The cycle loss is defined as $\mathcal{L}_{cyc,\mathcal{X}} = \mathbb{E}_{\boldsymbol{x} \sim P_{\mathcal{X}}}[d(\boldsymbol{x}, F_{\mathcal{Y} \to \mathcal{X}}(F_{\mathcal{X} \to \mathcal{Y}}(\boldsymbol{x})))]$, where $d$ is a distance function. The adversarial loss matches the real sample and the generated samples: $\mathcal{L}_{adv,\mathcal{X}} = \mathbb{E}_{\boldsymbol{x} \sim P_{\mathcal{X}}}[\mathcal{D}(\boldsymbol{x}, c_{\mathcal{X}})] - \mathbb{E}_{\boldsymbol{y} \sim P_{\mathcal{Y}}}[\mathcal{D}(F_{\mathcal{Y} \to \mathcal{X}}(\boldsymbol{y}), c_{\mathcal{X}})]$. The reconstruction loss helps the unsupervised encoder-decoder training: $\mathcal{L}_{rec,\mathcal{X}} = \mathbb{E}_{\boldsymbol{x} \sim P_{\mathcal{X}}}[d(\boldsymbol{x}, \mathcal{G}(E(\boldsymbol{x}, c_{\mathcal{X}}), c_{\mathcal{X}}))]$. The final objective sums three losses on both $\mathcal{X}$ and $\mathcal{Y}$, i.e. $\min_{\mathcal{G}} \max_{\mathcal{D}}[\mathcal{L}_{cyc,\mathcal{X}} + \mathcal{L}_{cyc,\mathcal{Y}} + \alpha \mathcal{L}_{adv,\mathcal{X}} + \alpha \mathcal{L}_{adv,\mathcal{Y}} + \beta \mathcal{L}_{rec,\mathcal{X}} + \beta \mathcal{L}_{rec,\mathcal{Y}}]$, where $\alpha, \beta$ are hyperparameters to balance the losses. In this task, NAGAN learns shared latent representations for two types of text, and then the alignment between two text spaces are strengthened by the cycle loss. We adapt NAGAN to this task with the non-autoregressive generator unchanged, where we further introduce an encoder and utilize a cross-entropy based discriminator (de Masson d'Autume et al., 2019). More details are described in Appendix A.6.1.

## 4 EXPERIMENTS

**Experiment Settings**. We test NAGAN on both synthetic and real data. The synthetic data (vocab size = 500, length = 20) is generated by an oracle Hidden Markov Model with fixed parameters. We do not use LSTM as the oracle model, because it may make the evaluation biased to the LSTM generators. We extract the text from COCO image caption (vocab size = 4,839, max length = 32) (Chen et al., 2015) and SNLI (vocab size = 42,981, max length = 40) (Bowman et al., 2015) dataset as the real data, which are adopted by Guo et al. (2018); Semeniuta1 et al. (2018).

**Dropout in NAGAN**. We utilize dropout (Srivastava et al., 2014) in our generator, which introduces noises for generating diverse samples and stabilizes GAN training (Isola et al., 2017; Sønderby et al., 2017). Note that dropout in GANs should also be applied in inference, which ensures that the generated samples are from the same distribution during GAN training. However, we find that varying the dropout rate in inference is in fact balancing the quality and the diversity of generated sentences, which will be shown in Section 4.1.

**Evaluation Metrics**. For the synthetic data, we calculate the oracle negative log-likelihood (Oracle NLL). For the real data, we adopt language model score (Caccia et al., 2020), $n$-gram based metrics (Shi et al., 2018), and Fréchet embedding distance (FED) (de Masson d'Autume et al., 2019). In these metrics, LM score and $\text{BLEU}_F$ measure quality, $\text{BLEU}_B$ measures diversity, $\text{BLEU}_{HA}$ and FED measure overall performance. The detailed definition are shown in Appendix A.4.1.

### 4.1 RESULTS ON UNCONDITIONAL TEXT GENERATION

We first test NAGAN on the synthetic data. The chosen baselines include a GRU model trained by MLE (Graves, 2013), two mainstream text GANs (Yu et al., 2017b; Guo et al., 2018), and ScratchGAN (de Masson d'Autume et al., 2019). As shown in Figure 2, NAGAN obtains lower Oracle NLL than the baselines, particularly than the other models without pretraining (solid lines). LeakGAN without pretraining is not shown because of too large scores (Oracle NLL=7.28).

On the real data, we test NAGAN against the MLE-trained GRU (Graves, 2013), Transformer (Vaswani et al., 2017), and the SOTA text GANs. The result is shown in Table 2. Our best model outperforms the pretrained baselines in LM Score and $\text{BLEU}_F$. With respect to diversity, MLE-trained models are superior to all text GANs except RelGAN in terms of $\text{BLEU}_{HA}$ on SNLI dataset. The result suggests that GAN training in these pretrained baselines, particularly with RL optimization (SeqGAN, LeakGAN, IRL), harms the diversity and even the overall performance, which was also reported previously (Caccia et al., 2020; Semeniuta1 et al., 2018).

Table 2: Generation performance on COCO and SNLI datasets. **Bold scores** indicate the best performance in all models while underline scores are the best in non-pretrained models.

| Model | COCO | | | | | SNLI | | | | |
|---|---|---|---|---|---|---|---|---|---|---|
| | LM Score ↓ | BLEU$_F$ ↑ | BLEU$_B$ ↑ | BLEU$_{HA}$ ↑ | FED ↓ | LM Score ↓ | BLEU$_F$ ↑ | BLEU$_B$ ↑ | BLEU$_{HA}$ ↑ | FED ↓ |
| *Pretrained Models:* | | | | | | | | | | |
| GRU (Graves, 2013) | 4.13 | 0.292 | **0.325** | 0.308 | **0.094** | 4.59 | 0.246 | **0.229** | 0.237 | **0.031** |
| Transformer (Vaswani et al., 2017) | 3.90 | 0.327 | 0.321 | **0.324** | **0.094** | 4.26 | 0.274 | 0.224 | 0.246 | 0.039 |
| SeqGAN (Yu et al., 2017b) | 4.03 | 0.298 | 0.285 | 0.291 | 0.108 | 3.96 | 0.293 | 0.176 | 0.220 | 0.113 |
| LeakGAN (Guo et al., 2018) | 3.81 | 0.321 | 0.239 | 0.273 | 0.324 | 4.40 | 0.259 | 0.200 | 0.226 | 0.054 |
| IRL (Shi et al., 2018) | 3.80 | 0.314 | 0.235 | 0.265 | 0.140 | 4.07 | 0.283 | 0.183 | 0.221 | 0.125 |
| RelGAN (Nie et al., 2019) | 3.71 | 0.347 | 0.288 | 0.315 | 0.164 | 3.76 | 0.332 | 0.215 | **0.261** | 0.086 |
| FMGAN (Chen et al., 2018) | 3.90 | 0.295 | 0.308 | 0.301 | 0.095 | 4.49 | 0.214 | 0.209 | 0.211 | 0.034 |
| *Non-pretrained Models:* | | | | | | | | | | |
| SeqGAN w/o pretrain | 9.06 | 0.019 | 0.021 | 0.020 | 0.812 | 6.99 | 0.148 | 0.065 | 0.090 | 0.262 |
| RelGAN w/o pretrain | 7.45 | 0.040 | 0.023 | 0.028 | 1.092 | 6.64 | 0.009 | 0.003 | 0.004 | 0.584 |
| FMGAN w/o pretrain | 8.83 | 0.004 | 0.027 | 0.007 | 0.928 | 9.65 | 0.002 | 0.001 | 0.002 | 0.601 |
| ScratchGAN | 4.13 | 0.278 | 0.251 | 0.264 | 0.127 | 4.57 | 0.237 | 0.192 | 0.212 | 0.062 |
| **NAGAN (dropout=0.25)** | 3.69 | 0.315 | 0.257 | 0.283 | 0.108 | 3.96 | 0.279 | 0.205 | 0.236 | 0.059 |
| **NAGAN (dropout=0.20)** | 3.54 | 0.342 | 0.256 | 0.293 | 0.111 | 3.78 | 0.310 | 0.205 | 0.246 | 0.067 |
| **NAGAN (dropout=0.15)** | 3.41 | 0.371 | 0.252 | 0.301 | 0.118 | 3.61 | 0.338 | 0.203 | 0.253 | 0.078 |
| **NAGAN (dropout=0.10)** | 3.30 | 0.391 | 0.245 | 0.301 | 0.128 | **3.48** | 0.363 | 0.196 | 0.254 | 0.096 |
| **NAGAN (dropout=0.00)** | **3.21** | **0.415** | 0.221 | 0.289 | 0.191 | **3.48** | **0.376** | 0.149 | 0.213 | 0.265 |

Among the non-pretrained models, NAGAN (dropout=0.25) outperforms previous text GANs in all metrics, where ScratchGAN (de Masson d'Autume et al., 2019) is the only baseline that can generate readable sentences. The other non-pretrained baselines fail to produce reasonable sentences, as their optimization methods are ineffective.

We test our models with different dropout rates in inference (where the training dropout rates remain unchanged as 0.25), and the result shows a smaller dropout rate can lead to higher quality. However, it also leads to less diversity (BLEU$_B$) and makes a larger gap between the distributions of the real sentences and the generated sentences (FED). It suggests that varying the dropout rate in inference is a method to balance the quality and diversity, where the larger noises introduced by the dropout encourage more diverse generated samples. We provide explanations in Appendix A.1.3.

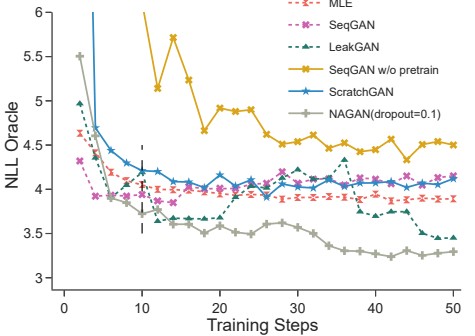

Figure 2: Training curves on synthetic data. The vertical dashed line denotes the end of pretraining for SeqGAN and LeakGAN.

Figure 3: Comparison to autoregressive (AR) generators on synthetic data of different lengths. Lower values are better.

## 4.2 WHY CAN NON-AUTOREGRESSIVE GAN WORK?

**Comparing Autoregressive (AR) and Non-autoregressive (NAR) Generator.** In Section 2, we mention that the AR generator can be an obstacle to effective optimization. To verify the conjecture, we replace NAGAN's NAR generator with an AR generator, while preserving the other parts unchanged. The non-differentiable issue should be tackled before optimizing AR generator with gradient-based methods, so we adopt two methods proposed in previous text GANs. The first one is called *Soft-Embedding* approximation (Chen et al., 2018). The generator predicts a word distribution at each position and obtains a soft word embedding weighted by the distribution, where the soft embedding is treated as the next input. The second one is called *Gumbel-Softmax* approximation (Kusner & Hernández-Lobato, 2016; Nie et al., 2019). In each step, a token is sampled from the word distribution with a reparameterized trick, where the straight-through estimator is also used for gradient estimation. We implement the AR generator using a Transformer with the same size of

Table 4: Performance with different discriminator architectures on COCO dataset.

| | LM Score ↓ | BLEU$_F$ ↑ | BLEU$_B$ ↑ | BLEU$_{HA}$ ↑ | FED ↓ |
|---|---|---|---|---|---|
| *Without Max Gradient Penalty:* | | | | | |
| Transformer | 4.62 | 0.111 | 0.051 | 0.070 | 0.734 |
| CNN | 5.52 | 0.202 | 0.188 | 0.195 | 0.167 |
| RNN | 5.27 | 0.187 | 0.110 | 0.138 | 0.446 |
| *With Max Gradient Penalty:* | | | | | |
| Transformer | **3.69** | **0.315** | **0.257** | **0.283** | **0.108** |
| CNN | 4.08 | 0.309 | 0.243 | 0.272 | 0.110 |
| RNN | 4.17 | 0.235 | 0.215 | 0.225 | 0.169 |

Table 5: Unsupervised decipherment. # and ∗ are reported in Chen et al. (2018) and He et al. (2020), respectively.

| Model | Accuracy on BrownW-200 |
|---|---|
| Gomez et al. (2018) | 75.7[#] |
| Chen et al. (2018) | 77.2[#] |
| **NAGAN** | **95.0** |
| | **BLEU-4 on WordSub** |
| Shen et al. (2017) | 50.8[∗] |
| Yang et al. (2018) | 49.3[∗] |
| He et al. (2020) | 78.4[∗] |
| **NAGAN** | **82.5** |

parameters with NAGAN. We also equip the generator with a latent variable, which is concatenated with the input at every step. Detailed implmentations are described in Appendix A.4.4.

We first compare NAGAN with these two methods on synthetic data of different lengths. As shown in Figure 3, NAGAN dominates the AR models on all lengths, and the AR models become worse rapidly as the length increases. Then we evaluate these models on SNLI dataset (max length = 40), where NAGAN remarkably outperforms the AR methods, as shown in Table 3. The results support

Table 3: Comparison to autoregressive (AR) generators on SNLI dataset.

| | LM Score ↓ | BLEU$_F$ ↑ | BLEU$_B$ ↑ | BLEU$_{HA}$ ↑ |
|---|---|---|---|---|
| NAGAN | **3.96** | **0.279** | **0.205** | **0.236** |
| AR(Soft) | 6.21 | 0.100 | 0.080 | 0.089 |
| AR(Gumbel) | 7.95 | 0.057 | 0.086 | 0.069 |

our claim that the feed-forward architecture benefits gradient-based optimization and allows GAN training from scratch possible. The AR generator suffers from ineffective optimization because of the non-differentiable issue at every step's input, especially with long sequences.

**Investigating Discriminator Architectures and Max Gradient Penalty (MaxGP).** To investigate the effects of the discriminator architectures and MaxGP technique, we utilize Transformer, CNN, and RNN as the discriminator, and evaluate them with and without MaxGP on COCO dataset. Detailed architectures are described in Appendix A.4.4. As shown in Table 4, MaxGP boosts the performance on all architectures, where the weakest RNN discriminator can realize effective optimization in the non-pretrained setting. Although not critical, the discriminator architectures indeed affect the results, suggesting that there is room for further improvement in the discriminator design.

## 4.3 APPLICATION I: MANIPULATING SENTENCES IN LATENT SPACE

In this task, we set dropout rate to zero in inference, where the generated text can be fully controlled by the latent variable. We evaluate NAGAN on COCO dataset with two metrics. *Success rate* denotes the proportion of edited sentences containing the desired word. For *overlap*, we first compute the ratio of the longest common subsequence's length to the maximal length of the two sentences, and then average the ratios over 100 successfully edited pairs.

We compare our proposed method (called GD, gradient descent) against the offset vector method (called OV) (Zhao et al., 2018). In GD, the moving step will be repeated for several iterations. For a fair comparison, we also allow OV to try the moving step $Z_s := Z_s + \beta\Delta$ for several iterations, where $\Delta$ is the offset vector. We also compare NAGAN against FMGAN (Chen et al., 2018), where the latter is equipped with latent variables but requires VAE pretraining. However, FMGAN cannot be applied to GD because it uses *Soft-Embedding* approximation to tackle the non-differentiable problem, which cannot provide gradient for latent variables during inference.

As presented in Table 6, NAGAN(OV) and FMGAN(OV) are comparable, but NAGAN(GD) achieves a higher success rate with better overlap scores, as it benefits from the direct gradient signals of the non-autoregressive generator. We also provide overall results over 100 random pairs of top-50 frequent words. The overall results are lower than the chosen cases because the random pairs may have different parts of speech, such as modifying *red* to *people*, making the task harder.

Table 6: Results for sentence manipulation (modifying a sentence containing a **source word** to the one containing a *target word*). *OV* = Offset Vector; *GD* = Gradient Descent (Ours).

| | FMGAN(OV) | | NAGAN(OV) | | NAGAN(GD) | | |
|---|---|---|---|---|---|---|---|
| src./tar. words | Succ. | Overlap | Succ. | Overlap | Succ. | Overlap | Cases of Sentence Manipulation by NAGAN(GD) |
| black
white | 0.59 | 0.32 | 0.72 | 0.39 | **0.99** | **0.75** | A **black** cat laying outside in front of a large table.
A *white* cat laying outside in front of a large table. |
| white
black | 0.80 | 0.37 | 0.40 | 0.47 | **0.96** | **0.72** | A **white** cat sitting on the top of a large chair.
A *black* cat sitting in the middle of a large chair. |
| standing
sitting | 0.81 | 0.30 | 0.74 | 0.46 | **1.00** | **0.71** | Two people are **standing** next to a window in an umbrella.
A man is *sitting* next to a window in an umbrella. |
| sitting
standing | 0.47 | 0.33 | 0.75 | 0.45 | **0.98** | **0.64** | A man **sitting** in front of a window on a lake.
A man *standing* in front of a boat on a lake. |
| Overall | 0.62 | 0.24 | 0.54 | 0.33 | **0.76** | **0.47** | |

## 4.4 APPLICATION II: UNSUPERVISED DECIPHERMENT

We use Word Substitution dataset (Yang et al., 2018) and BrownW-200 (Gomez et al., 2018) for the task, where the text is encrypted by the substitution and the Vigenère cipher, respectively. Following previous work, WordSub is reported with BLEU-4, and BrownW-200 is reported with accuracy.

As shown in Table 5, NAGAN significantly outperforms existing methods. Compared with AR models, the NAR generator facilitates effective optimization for the cycle loss and the adversarial loss, where AR generators are facing the same optimizing problem as discussed in Section 4.2. The gradient of both losses can be back-propagated through our feed-forward generator smoothly, leading to a better alignment between two text spaces and thus a better decipher result.

## 5 CONCLUSIONS

We present a novel text GAN, named NAGAN, which incorporates a non-autoregressive generator to facilitate efficient gradient-based training from scratch and the use of latent variables. NAGAN adopts a novel formulation for adversarial text generation, which connects itself with image GANs and can potentially benefit from the success of image generation. As a preliminary work of adversarial non-autoregressive text generation, NAGAN shows promising results on generating diverse sentences without conditions, which may suggest a direction for solving the *multi-modality* problem (Gu et al., 2018) and can be applied to general sequence-to-sequence generation as future work.

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

## A  APPENDIX

### A.1  MORE DISCUSSIONS

#### A.1.1  DEGENERATION OF AUTOREGRESSIVE MODELS WITH LATENT VARIABLE

An autoregressive model with latent variables can be defined as follows (Bowman et al., 2016; Lucas & Verbeek, 2018):

$$P_{\mathcal{G}}(\boldsymbol{x}) = \mathbb{E}_{\boldsymbol{z} \sim p(\boldsymbol{z})} \prod_{i=1}^{L} P_{\mathcal{G}}(\boldsymbol{x}_i | \boldsymbol{x}_{<i}, \boldsymbol{z}). \tag{14}$$

It can degenerate into a language model, where the latent variable $\boldsymbol{z}$ is ignored when generating $\boldsymbol{x}$, i.e., mutual information $I_{\mathcal{G}}(\boldsymbol{x}, \boldsymbol{z}) = 0$. This problem has been found in Variational Autoencoders (VAEs), known as the *KL vanishing* problem (Bowman et al., 2016). However, the reconstruction loss in VAEs optimizes the lower bound of $I_{\mathcal{G}}(\boldsymbol{x}, \boldsymbol{z})$, proven by Li et al. (2017) in Corollary 3:

$$I_{\mathcal{G}}(\boldsymbol{x}, \boldsymbol{z}) \geq \mathbb{E}_{\boldsymbol{x} \sim p_{data}} \left[ \mathbb{E}_{\boldsymbol{z} \sim q(\boldsymbol{z}|\boldsymbol{x})} [\log P_{\mathcal{G}}(\boldsymbol{x}|\boldsymbol{z})] \right] + H_{p_{data}}(\boldsymbol{x}) \tag{15}$$

$$= -\mathcal{L}_{recon} + constant \tag{16}$$

where $H$ means entropy, and $q$ is the approximate posterior distribution. Therefore, optimizing $\mathcal{L}_{recon}$ avoids the degeneration, making VAEs able to learn latent representations.

However, text GANs only use an adversarial loss, which merely matches $P_{\mathcal{G}}$ and $P_{real}$. We can easily construct an ideal generator defined as $P_{\mathcal{G}}(\boldsymbol{x}_i | \boldsymbol{x}_{<i}, \boldsymbol{z}) = P_{real}(\boldsymbol{x}_i | \boldsymbol{x}_{<i})$, where $P_{\mathcal{G}}$ perfectly fits $P_{real}$, but $I_{\mathcal{G}}(\boldsymbol{x}, \boldsymbol{z})$ equals 0. We can conclude that the text GAN with an autoregressive generator is not guaranteed to learn latent representations.

Things are different if $\mathcal{G}$ is a deterministic function, where the generator $\mathcal{G}$ is forced to use $\boldsymbol{z}$ if $P_{\mathcal{G}}$ can fit $P_{real}$ well. To explain the statement, we present another lower bound of mutual information. Let $\mathcal{G}_{\theta}$ be an generator with parameters $\theta$, and $p(\boldsymbol{z})$ be the prior distribution of $\boldsymbol{z}$, we have:

$$I_{\mathcal{G}_{\theta}}(\boldsymbol{x}, \boldsymbol{z}) = \mathbb{E}_{\boldsymbol{z} \sim p(\boldsymbol{z})} \left[ D_{\mathrm{KL}} \left( P_{\mathcal{G}_{\theta}}(\boldsymbol{x}|\boldsymbol{z}) \| P_{\mathcal{G}_{\theta}}(\boldsymbol{x}) \right) \right]$$

$$= \mathbb{E}_{\boldsymbol{z} \sim p(\boldsymbol{z})} \left[ D_{\mathrm{KL}} \left( P_{\mathcal{G}_{\theta}}(\boldsymbol{x}|\boldsymbol{z}) \| P_{real}(\boldsymbol{x}) \right) + \mathbb{E}_{\boldsymbol{x} \sim P_{\mathcal{G}_{\theta}}(\boldsymbol{x}|\boldsymbol{z})} \left[ \frac{\log P_{real}(\boldsymbol{x})}{\log P_{\mathcal{G}_{\theta}}(\boldsymbol{x})} \right] \right]$$

$$= \mathbb{E}_{\boldsymbol{z} \sim p(\boldsymbol{z})} \left[ D_{\mathrm{KL}} \left( P_{\mathcal{G}_{\theta}}(\boldsymbol{x}|\boldsymbol{z}) \| P_{real}(\boldsymbol{x}) \right) \right] - D_{\mathrm{KL}} \left( P_{\mathcal{G}_{\theta}}(\boldsymbol{x}) \| P_{real}(\boldsymbol{x}) \right) \tag{17}$$

$$\geq \min_{\boldsymbol{z}^*, \theta^*} \left[ D_{\mathrm{KL}} \left( P_{\mathcal{G}_{\theta^*}}(\boldsymbol{x}|\boldsymbol{z}^*) \| P_{real}(\boldsymbol{x}) \right) \right] - D_{\mathrm{KL}} \left( P_{\mathcal{G}_{\theta}}(\boldsymbol{x}) \| P_{real}(\boldsymbol{x}) \right) \tag{18}$$

In Eq (17), we relax the first term by fixing $\boldsymbol{z}$ and $\theta$ to minimize the KL divergence. In the first term of Eq (18), $\boldsymbol{z}^*$ is a constant, and thus $P_{\mathcal{G}_{\theta^*}}(\boldsymbol{x}|\boldsymbol{z}^*)$ can be regarded as a generator with a fixed input, which ignores the latent variable. In the second term, $P_{\mathcal{G}_{\theta}}(\boldsymbol{x})$ is the expectation of $P_{\mathcal{G}_{\theta}}(\boldsymbol{x}|\boldsymbol{z})$ over $p(\boldsymbol{z})$, which represents the model distribution considering latent variables. Then the first term indicates the best KL divergence if $\mathcal{G}$ ignores the latent variable, and the second term indicates the KL divergence when $\mathcal{G}_{\theta}$ uses the latent variable. Therefore, $I_{\mathcal{G}_{\theta}}(\boldsymbol{x}, \boldsymbol{z})$ is bounded by the performance gap between the generators with and without the latent variable.

If $\mathcal{G}$ is a deterministic function, $P_{\mathcal{G}_{\theta^*}}(\boldsymbol{x}|\boldsymbol{z}^*)$ must be a one-point distribution[5], which is far from fitting the real data. As long as $\mathcal{G}_{\theta}$ can be effectively optimized, the second term of Eq (18) should

---

[5] $P_{\mathcal{G}_{\theta^*}}(\boldsymbol{x}|\boldsymbol{z}^*)$ is equal to 1 when $\boldsymbol{x} = \mathcal{G}_{\theta^*}(\boldsymbol{z}^*)$ and equal to 0 otherwise.

be smaller than the first term. It indicates $I_{\mathcal{G}_\theta}(\boldsymbol{x}, \boldsymbol{z}) > 0$, and $\mathcal{G}_\theta$ cannot ignore the latent variable. Moreover, when $\mathcal{G}_\theta$ is optimized to approach the real distribution, the lower bound of the mutual information becomes larger, which usually indicates better representations (Chen et al., 2016; Hjelm et al., 2019).

### A.1.2  SENTENCE LENGTH IN NON-AUTOREGRESSIVE GENERATORS

Autoregressive text generators use a special token (e.g., <eos>) to determine the length, but most non-autoregressive text generators require the sentence length before the parallel generation. Strictly speaking, a non-autoregressive text generator is a conditional generator, which generates the text conditioned on the length $L$ and the input $C$, i.e., $\boldsymbol{x} = \mathcal{G}(C, L)$. Therefore, the generation process is split into two steps: determine the sentence length, and then generate sentences according to the length. We use the non-autoregressive generator only in the second step, where the first step can vary in different tasks.

In machine translation, non-autoregressive text generator usually uses a length predictor to predict the target length from the input sentence (Gu et al., 2018), which is trained separately from the non-autoregressive generator. There is also an alternative method called Length-parallel Decoding (Shu et al., 2020), where the generator tries all possible lengths (by enumerating a pre-specified range) and generates a set of sentences, and then select the best one using an external ranking model.

In unconditional text generation, we aim to match $P_{\mathcal{G}}(\boldsymbol{x})$ with the real distribution. Regarding the length $L$ as another latent variable, $P_{\mathcal{G}}(\boldsymbol{x})$ can be formulated as a sampling process: $L \sim p(L), \boldsymbol{z} \sim p(\boldsymbol{z}|L), \boldsymbol{x} = \mathcal{G}(\boldsymbol{Z}, L)$. We intuitively estimate $p(L)$ by the sentence length distribution of the training data, which helps $P_{\mathcal{G}}(\boldsymbol{x})$ approach $P_{real}(\boldsymbol{x})$ well.

In the unsupervised word decipherment application, we have already known the target length should be equal to the source length, so we can simply obtain the target length. However, it is theoretically possible to learn a length predictor $P_C(L|\boldsymbol{y})$ to predict the target length, where $\boldsymbol{y}$ is the ciphertext. Then the decipher model $F_{\mathcal{Y} \to \mathcal{X}}$ can be formulated as a sampling process: $\boldsymbol{Z} = E(\boldsymbol{y}, c_{\mathcal{Y}}), L \sim P_C(L|\boldsymbol{y}), \boldsymbol{x} = \mathcal{G}(\boldsymbol{Z}, c_{\mathcal{X}}, L)$. The classifier $P_C$ can be trained to minimize the final objective described in Section 3.4, by reinforcement learning or other optimization methods.

### A.1.3  HOW DROPOUT WORK IN GANS

This subsection is added during the discussion.

Dropout is a common trick in training image GANs, but it is more than a simple regularizer when used in GANs' generators. We provide two explanations about why dropout can stabilize the training process and balance fluency and diversity.

First, dropout can be regarded as noises to diversify the generated samples. GANs training usually suffers from the mode collapse problem, which can be alleviated by keeping the diversity of the generated samples. We refer the reader to the *instance noise* trick (Sønderby et al., 2017), which is also a method that keeps the diversity. The author claim that the diverse samples can stabilize the gradient flows by avoiding the situation that $P_{\mathcal{G}}$ and $P_{data}$ have disjoint supports.

When removing the dropout in NAGAN, we can observe spiky losses and unstable training process. However, it still outperforms some baselines without MLE pretraining. The results of no dropout training is demonstrated in Table 11.

Second, dropout can be regarded as a method of introducing extra latent variables. Unlike the common networks that learn to produce the same output when dropout masks change, it is shown that random noises introduced in the intermediate layer of GANs' generator will affect the generated samples (Karras et al., 2019). We can explicitly take these random noises as latent variables, where different noises will be mapped to different samples in the GANs training. In our case, the latent variables are the dropout masks.

A linear layer with dropout can be formulated as follows:

$$\boldsymbol{y} = \frac{1}{1-p} \boldsymbol{W}(\boldsymbol{x} \otimes \boldsymbol{m}), \tag{19}$$

where $\boldsymbol{x}$ is the input vector, $\boldsymbol{y}$ is the output vector, $\boldsymbol{W}$ is the weight matrix, $\boldsymbol{m}$ is the dropout mask, $p$ is the dropout rate, and $\otimes$ is element-wise multiplication. Taking $\boldsymbol{m}$ as a random variable, we can obtain the mean and the variance of $\boldsymbol{y}$.

$$
\begin{aligned}
\mathbb{E}[\boldsymbol{y}] &= \frac{1}{1-p} \sum_{\boldsymbol{m} \in \{0,1\}^N} \boldsymbol{W}(\boldsymbol{x} \otimes \boldsymbol{m}) P(\boldsymbol{m}) \\
&= \frac{1}{1-p} \sum_{i=1}^N \boldsymbol{W_i} \sum_{m_i \in \{0,1\}} (x_i m_i P(m_i)) \\
&= \frac{1}{1-p} \sum_{i=1}^N \boldsymbol{W_i} x_i (1-p) = \boldsymbol{W}\boldsymbol{x}
\end{aligned}
\tag{20}
$$

$$
\begin{aligned}
\mathrm{Var}[\boldsymbol{y}] &= \mathbb{E}[\boldsymbol{y}^2] - \mathbb{E}^2[\boldsymbol{y}] \\
&= \frac{1}{(1-p)^2} \sum_{i=1}^N \boldsymbol{W_i} \sum_{m_i \in \{0,1\}} (x_i^2 m_i^2 P(m_i)) - (\boldsymbol{W}\boldsymbol{x})^2 \\
&= (\frac{1}{1-p} - 1)(\boldsymbol{W}\boldsymbol{x})^2
\end{aligned}
\tag{21}
$$

$N$ is the dimension of $\boldsymbol{x}$, $\boldsymbol{W}_i$ is the $i$-th column of $\boldsymbol{W}_i$. When $p = 0$, the variance of $\boldsymbol{y}$ is 0. For a larger $p$, the mean keeps unchanged but the variance becomes larger.

In the GANs training and inference, we usually keep the distribution of dropout masks, i.e., the dropout rate, unchanged (Isola et al., 2017), so that the generator's distribution remains unchanged. However, if we reduce the dropout rate in inference, the distribution will be more concentrated, bringing samples with higher quality but lower diversity.

This phenomenon is not significant in autoregressive text GANs. One reason is that the implicit generative model uses a deterministic generator, where the generated sample is fully determined by the latent variables (including $\boldsymbol{Z}$ and dropout masks). However, the autoregressive generator samples from word distribution at each step, which brings more diversity than a non-autoregressive generator, so the effect of dropout will be weakened.

## A.2 Exploring the Latent Space and Generation Process

==This section is added during the discussion.==

For a better understanding of the learned latent space and the non-autoregressive generation process, we further conduct four experiments on the COCO dataset.

### A.2.1 Sentence Interpolation

==This subsection is added during the discussion.==

We set $L = 12$ and randomly sample two sequences of latent variables $\boldsymbol{Z}_1, \boldsymbol{Z}_2$ from $\mathcal{N}(\boldsymbol{0}, \boldsymbol{I})$ independently. Then we obtain the linearly interpolated latent variables $\boldsymbol{Z}_\lambda = (1 - \lambda)\boldsymbol{Z}_0 + \lambda \boldsymbol{Z}_1$ and translate them to the text space. Cases are shown in Table 7.

### A.2.2 Smoothness of Sentences with Different Lengths

==This subsection is added during the discussion.==

We randomly sample a sequence of latent variables $\boldsymbol{Z}$ and then concatenate a new sampled $\boldsymbol{z}_{L+1}$ to the end of $\boldsymbol{Z}$. We translate two sequences of latent variables to text space and find that the two sentences are very similar, except the length differs. We repeat the step and generate sentences with length from 10 to 17. We also try to concatenate a new sampled variable to the front of the sequence. The generated sentences are shown in Table 8.

### A.2.3 Connection between $\mathbf{Z}_i$ and $\mathbf{O}_i$

==This subsection is added during the discussion.==

Table 7: Cases of sentence interpolation. Gray words indicate unchanged parts compared with the preceding sentence.

| $\lambda$ | Generated Sentences |
|---|---|
| 0.0 | A picture of a dog laying on posts in the forest . |
| 0.1 | A picture of a person laying a runway in the ocean. |
| 0.2 | A picture of an person on a motorcycle in the ocean. |
| 0.3 | A picture of an elephant riding a surfboard in the ocean. |
| 0.4 | A picture of an elephant riding a surfboard in the ocean. |
| 0.5 | A picture of the ocean while some waves in the ocean. |
| 0.6 | A picture of the ocean while some surfboards in the ocean. |
| 0.7 | A picture of the ocean that being sailing in the water. |
| 0.8 | A picture of the ocean that is walking in the water. |
| 0.9 | A man in the ocean that is surfing in the ocean. |
| 1.0 | A man in a suit that is surfing in the ocean. |
| 0.0 | A big teddy bear laying on a rock in a field. |
| 0.1 | A herd teddy bear laying on a rock in a field. |
| 0.2 | A herd of sheep laying on a rock in a field. |
| 0.3 | A herd of cattle grazing on a rock in a field. |
| 0.4 | A group of cattle standing on a motorcycle in a field. |
| 0.5 | A group of cattle standing on a motorcycle in the field. |
| 0.6 | A group of people standing on a motorcycle in the park. |
| 0.7 | A group of people standing on a horse in the rain. |
| 0.8 | A group of people standing riding a horse in the rain. |
| 0.9 | A group of young man riding a horse in the rain. |
| 1.0 | A group of young man riding a horse in the rain. |
| 0.0 | A man in a suit riding a skateboard in the snow. |
| 0.1 | A man in a suit riding some surfboards in the snow. |
| 0.2 | A man in a suit riding some surfboards in the ocean. |
| 0.3 | A picture of a person riding some surfboards in the rain. |
| 0.4 | A picture of a giraffe trying to someone in the rain. |
| 0.5 | A group of cattle standing next to someone in the rain. |
| 0.6 | A group of people standing next to someone in the distance. |
| 0.7 | A group of giraffes standing next to someone in the pasture . |
| 0.8 | A group of giraffes standing next to someone in the pasture. |
| 0.9 | A group of giraffes standing next to someone in the pasture. |
| 1.0 | A rows of giraffes stand next to other in the pasture. |

Table 8: Smoothness of sentences with different lengths. We convert one sentence to the next one by adding a new sampled latent variable to the end (or the front) of the latent variable sequence. Gray words indicate unchanged parts compared with the preceding sentence.

| Length | Added to the end |
|---|---|
| 10 | A pile of fresh fries, broccoli and mashed . |
| 11 | A pile of broccoli and a sandwich in mashed fries . |
| 12 | A slice of bread with a white sandwich on their fries . |
| 13 | A white white plate with a white cream arranged on a plate . |
| 14 | A white plate with chicken sized bunches of broccoli , and french fries . |
| 15 | A white plate with various meat and mashed broccoli , broccoli and french fries . |
| 16 | A white plate with meat peppers and mashed potatoes , broccoli , and french fries . |
| 17 | A white plate with chicken peppers and mashed potatoes , broccoli , and french fries nearby . |

| Length | Added to the front |
|---|---|
| 10 | A pile of bread on a plate in pan . |
| 11 | A plate is pie on a plate in a pan . |
| 12 | A slice of bread with sliced toppings sauce on a plate . |
| 13 | A white plate with sliced toppings sitting on a dinner white plate . |
| 14 | A plate of fresh fruits sitting on a plate on a white pan . |
| 15 | A pile of fruits and vegetables on a plate with broccoli and a fork . |
| 16 | A pile of broccoli and vegetables on top with a sandwich and french fries nearby . |
| 17 | A pile of broccoli and vegetables on top with a sandwich , and a plate nearby . |

As the length of $\boldsymbol{Z}$ is always equal to the length of generated sentences, it seems natural if $\boldsymbol{z}_i$ is highly related to $\boldsymbol{o}_i$. However, the following experiment negates the hypothesis.

We first sample a sequence of latent variables $\boldsymbol{Z}$ ($L = 10$), and translate it to sentence $\boldsymbol{O}$. Then we random choose a position $i$ and replace $\boldsymbol{z}_i$ by an new sampled latent variable from $\mathcal{N}(\boldsymbol{0}, \boldsymbol{I})$. The new sequence is denoted by $\boldsymbol{Z}'$ and translated to a new sentence $\boldsymbol{O}'$. We repeat the trials for 1,600 times and count the probability $P(i, j)$ that $\boldsymbol{o}_j$ changed when modifying $\boldsymbol{z}_i$. If the hypothesis is true, high probabilities should be observed in the diagonal of the matrix $P$. However, $P(i, j)$ is highly correlated with the position $i$ regardless of $j$, as shown in Figure 4. We can conclude that there is no strong connection between $x_i$ and $z_i$, and each latent variable plays a similar role in generating $x_i$.

This property is not desired and can make NAGAN less interpretable. However, we think that adding regularizers will be a good way to make the tokens more correlated to nearby latent variables, which is left for future work.

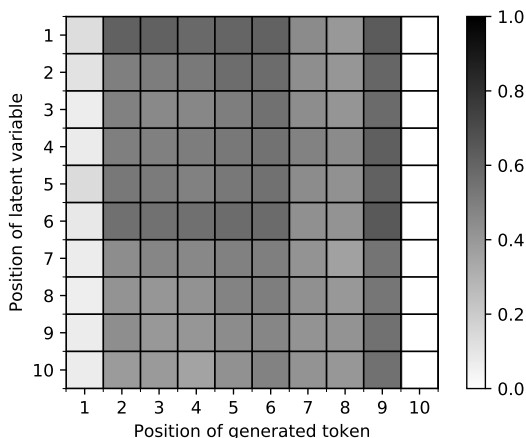

Figure 4: Connection between tokens and latent variables at different positions. The element in the $i$-th row and the $j$-th column is the probability that the token $\boldsymbol{x}_j$ changes when modifying the latent variable $\boldsymbol{z}_i$. The last token is a full stop and never changed.

### A.2.4 INVESTIGATING GENERATION PROCESS

This subsection is added during the discussion.

Autoregressive generators always generate tokens from left to right, but non-autoregressive generators produce the whole sequence in parallel. We investigate the generation process of NAGAN's generator and find some interesting properties: The non-autoregressive generator will gradually determine the hidden states as the data flows through the Transformer layers; It tends to determine some of the tokens first and then the others.

We collect the hidden states of each Transformer's layer and calculate the cosine similarities between the intermediate hidden states and the last outputs. To be specific, $sim_{t,l} = cos(\boldsymbol{h}_{t,l}, \boldsymbol{h}_{t,n})$, where $\boldsymbol{h}_{t,l}$ is the output of the $l$-th layer at the $t$-th position, and $L$ is the number of layers. In our experiment, $L = 5$ and $1 \leq t \leq 10$.

As shown in Figure 5, the similarities at each position are usually slowly increased from the first layer to the last layer, which means the generator gradually determine their hidden states. Some tokens at the front ($t = 1$), in the middle ($4 \leq t \leq 7$), and at the end ($t = 10$) will be determined before the 3rd layer, and the other tokens will be determined in the last two layers. The tokens at the front are usually definite or indefinite articles, and the tokens at the end are full stops. The tokens in the middle are usually prepositions (e.g., *in*, *top*, and *next*), which split the whole sentences in two halves. This strategy may make the generation of the remained tokens easier, because it is similar to a masked language model when the near context is given.

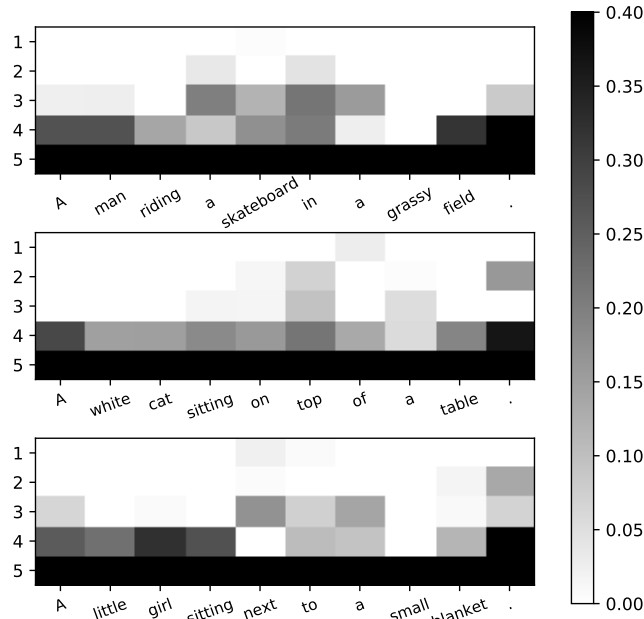

Figure 5: Cases of cosine similarities between intermediate hidden states and the last outputs. The $i$-th row indicates the hidden states of the $i$-th Transformer layer ($1 \leq i \leq 5$).

### A.3    IMPLEMENTATION DETAILS

Here we introduce the implementation details of NAGAN. The codes are available in the supplementary material.

#### A.3.1    GENERATOR

The generator's Transformer is based on the architecture of the non-autoregressive text decoder[6] (Gu et al., 2018), which contains $n\_layers$ Transformer blocks. The attention layers attending to the source sentence are replaced by attention layers attending to the latent variable.

In the training stage, we introduce Gumbel noises when obtaining $s_t$. Introducing noises is a common technique used in image GANs (Zhao et al., 2017; Sønderby et al., 2017), where noises can encourage the model to generate diverse samples and avoid *mode-collapse* problem in the early training stage. To be specific, $s_t = \text{MLP}(h_t) + g_t$, where $g_t$ is sampled from the standard Gumbel distribution. We use Gumbel noises rather than the other noises (e.g., Gaussian noises) because they are more suitable for categorical distribution (Jang et al., 2017). Different from the Gumbel-Softmax tricks, we do not use the noise or sample from the categorical distribution in inference.

As mentioned in Section 3.1, the straight-through estimator is only used when obtaining gradient. We provide an alternative description for the implementation as follows:

$$o'_t = \text{onehot}(\arg\max_v(s_{t,v})), \tag{22}$$

$$o''_t = \text{softmax}(s_t/\tau), \tag{23}$$

$$o_t = \text{stop\_gradient}(o'_t - o''_t) + o''_t. \tag{24}$$

where $s_{t,v}$ is the $v$-th dimension of $s_t$. It means that we use $o_t = o'_t$ in the forward pass, and $o_t = o''_t + constant$ when obtaining gradient.

---

[6]Our implementation is based on https://github.com/salesforce/nonauto-nmt.

### A.3.2 DISCRIMINATOR

The discriminator's Transformer has the same architecture as the encoder[6] in Gu et al. (2018), which contains $n\_layers$ Transformer blocks.

**Max Gradient Penalty**. Following Zhou et al. (2019), we calculate a max sampled gradient as follows:

$$\mathcal{L}_{GP} = \max_{i=1,2,\cdots,n} \left\| \frac{\partial \mathcal{D}(\boldsymbol{O}'_i)}{\partial \boldsymbol{O}'_i} \right\|^2, \tag{25}$$

where $n$ denotes the batch size. The intermediate sequences $\boldsymbol{O}'_i$ are sampled in the following way: first, sample a sentence from the real data, whose one-hot representation is $\boldsymbol{O}_{ri}$; second, generate fake sample $\boldsymbol{O}_{fi}$ with the same length of $\boldsymbol{O}_{ri}$. Finally,

$$\boldsymbol{O}'_i = \lambda_i \boldsymbol{O}_{ri} + (1 - \lambda_i) \boldsymbol{O}_{fi} \tag{26}$$

where $\lambda_i$ is sampled from $[0, 1]$ uniformly. Note that $\boldsymbol{O}'_i$ may not be a one-hot sequence and does not represent a discrete sentence. However, it is not a problem because it is only used to regularize the discriminator.

**Score Regularizer**. We also adopt a L2-regularizer on the predicted score $\mathcal{D}(\boldsymbol{O})$, which is proposed to stabilize GAN training (Xu et al., 2019). It can be formulated as follows:

$$\mathcal{L}_{SR} = \left\| \mathcal{D}(\boldsymbol{O}) \right\|^2. \tag{27}$$

### A.3.3 TRAINING

The training procedure is presented in Algorithm 1.

---

**Algorithm 1** Training Procedure of NAGAN

---

**Require:**    Max Training Steps: $iters$    Batch Size: $n$
1: **for** $T = 1, 2, \cdots, iters$ **do**
    ▷ *Training the discriminator.*
2:    Sample a batch of real data $\boldsymbol{O}_{ri} (1 \leq i \leq n)$, whose length is $L_i$.
3:    Sample a batch of latent variables $\boldsymbol{Z}_i (1 \leq i \leq n)$, where $\boldsymbol{Z}_i \in \mathbb{R}^{\text{Z\_dim} \times L_i}$.
4:    Translate the latent variables $\boldsymbol{Z}_i$ to generated sentences $\boldsymbol{O}_{fi}$ via the generator $\mathcal{G}$, for $1 \leq i \leq n$.
5:    Calculate $L_{GP}$ via Eq (25).
6:    Update the discriminator $\mathcal{D}$ via $\mathcal{L}'_{\mathcal{D}} = \mathcal{L}_{\mathcal{D}} + \lambda(\mathcal{L}_{GP} + \mathcal{L}_{SR})$ (See Eq (12) for $\mathcal{L}_{\mathcal{D}}$).
    ▷ *Training the generator.*
7:    Sample a new batch of latent variables $\boldsymbol{Z}_i$ similar to (3).
8:    Update the generator $\mathcal{G}$ via Eq (13).
9: **end for**

---

**Parameter Averaging**. Some studies (Mescheder et al., 2017; Nagarajan & Kolter, 2017; Mescheder et al., 2018) analyze the training dynamics of GANs and show that the generator does not converge to a point but becomes a periodic function around the Nash equilibrium point. Therefore we adopt a common method used in image GAN, called parameter averaging (Yazici et al., 2019).

Suppose the generator's parameter at step $t$ is $\theta_t$. Then, the exponential moving average is defined as

$$\hat{\theta}_t = \gamma \hat{\theta}_{t-1} + (1 - \gamma)\theta_t, \tag{28}$$

where $\gamma$ is often a number slightly smaller than 1. It is noticeable that $\hat{\theta}_t$ has much smaller variance than $\theta_t$, and $\hat{\theta}_t$ is then used for the evaluation.

### A.4 EXPERIMENT SETTINGS

### A.4.1 EVALUATION METRICS

**Oracle NLL**. For synthetic data, we calculate the negative log-likelihood (NLL) per token of the oracle model evaluated on 5,000 generated sentences. We do not use perplexity because the likelihood of implicit generative models is intractable.

**Language Model Scores**.   Language model score (Ke et al., 2019; Caccia et al., 2020) is the NLL of a language model trained on the test set and evaluated on 5,000 generated sentences. As suggested by de Masson d'Autume et al. (2019), this score may favor the models which have the same architecture of the language model. Thus we use a 4-gram language model with Kneser–Ney smoothing (Ney et al., 1994) instead of using RNN language models.

$n$**-gram Based Metrics**.   To evaluate both the quality and the diversity (Shi et al., 2018), we adopt three BLEU scores: Forward BLEU ($BLEU_F$), Backward BLEU ($BLEU_B$), and their harmonic mean ($BLEU_{HA}$). Forward BLEU measures quality, which uses the test set as references, and evaluates 5,000 generated sentences with the BLEU score. Backward BLEU measures diversity, which swaps the roles of the test set and the generated sentences. We use the BLEU-5 score in our experiments.

Some previous work adopts self-BLEU (Zhu et al., 2018) for evaluating diversity, which calculates the BLEU score of each generated sentence using the other generated sentences as references. The metric suffers from *meaningless diversity*, where a model that randomly generates tokens can achieve perfect self-BLEU. We use Backward BLEU to measure diversity, which is symmetric with Forward BLEU and avoids the *meaningless diversity* problem.

**Fréchet Embedding Distance (FED)**.   FED (de Masson d'Autume et al., 2019) is a text version of Fréchet Inception Distance (Heusel et al., 2017), which has been widely used in image GANs. It measures the similarity between the representation distributions of the test data and the generated data, where we use 5,000 generated sentences as the generated data. Sentences are represented with vectors by Universal Sentence Encoder[7] (Cer et al., 2018).

### A.4.2   SYNTHETIC DATA

The synthetic data are generated by an HMM model[8] with 100 hidden states. A randomly initialized HMM may have a flat distribution over all sequences, which is remarkably different from natural language text, so we fit the HMM on 5,000 real sentences from the COCO dataset before generating the synthetic data. However, the generated synthetic data are still different from the COCO dataset, where the synthetic sentences have a fixed length (length=20 in Figure 2) and a smaller vocabulary of 500 tokens. The parameters of the oracle HMM will be released. We generate 50,000 sentences as the training set. We do not use a validation set or test set, because the Oracle NLL does not need real samples for evaluation. The Oracle NLL is always averaged over 5,000 generated sentences.

The baselines with the RNN architecture are using LSTM with 128 cells or GRU with 256 cells. The MLE baseline is a 256-dim GRU. The other hyper-parameters of baselines are setting according to the official implementation[9]. The hyper-parameters of NAGAN are shown in Table 9.

In Table 10, we provide the best Oracle NLLs as a supplement of Figure 2. Note that one training step in Figure 2 may be different across models. For MLE, ScratchGAN, NAGAN, and the pretraining part of SeqGAN and LeakGAN, one training step means one epoch training. For the GAN training part of SeqGAN and LeakGAN, one training step contains several updating steps in the generator and the discriminator, which is consistent with the original implementations.

### A.4.3   REAL DATA

For the COCO dataset (Chen et al., 2015), we follow previous work (Guo et al., 2018; Shi et al., 2018; Ke et al., 2019) and use the image captions as the real sentences, where the images are discarded. The dataset is processed by Shi et al. (2018) [10], which contains 80,000 samples in the training set, 5,000 samples in the test set, and 4,838 words in the vocabulary. However, the dataset does not contain a validation set, so we randomly choose 5,000 samples from the training set as our validation set.

For the SNLI dataset (Bowman et al., 2015), we follow previous work (Semeniuta1 et al., 2018; Zhao et al., 2018) to extract the sentences, where the connections between the paired sentences are ignored. The SNLI dataset contains 714,667 samples in the training set, 5,000 samples in both the validation and test set, and 42,981 words in the vocabulary. We trim long sentences to the first 40 tokens.

---

[7]The pretrained model can be downloaded at https://tfhub.dev/google/universal-sentence-encoder/3

[8]We use the implementation of https://github.com/hmmlearn/hmmlearn.

[9]https://github.com/geek-ai/Texygen

[10]https://github.com/FudanNLP/Irl_gen.

Table 9: Hyper-parameters of NAGAN on the synthetic data and the real data.

| Hyperparameter | Value |
|---|---|
| MaxGP Weight $\lambda$ (Synthetic / Real) | 0.1 / 1 |
| Softmax Temperature $\tau$ | 1 |
| Parameter Averaging $\gamma$ | 0.9999 |
| Batch Size (Synthetic / Real) | 128 / 64 |
| # Transformer Blocks ($n\_layers$) | 5 |
| # Transformer Heads | 4 |
| Dim. of Transformer ($tf\_dim$) | 128 |
| Dim. of Latent Variable ($Z\_dim$) | 64 |
| Dim. of Word Embedding ($E\_dim$) | 300 |
| Droprate in Training | 0.25 |
| L2 Regularizer ($\mathcal{D}$ / $\mathcal{G}$) | 0.05 / 0 |
| Optimization method | RAdam (Liu et al., 2020) |
| Learning rate ($\mathcal{D}$ / $\mathcal{G}$) | 5e-4 / 2e-4 |
| $\beta_1, \beta_2$ in RAdam | 0.5, 0.9 |

Table 10: Results on the synthetic data. NA-GAN are tagged with droprates in inference (Training droprate is always 0.25).

| Model | Oracle NLL |
|---|---|
| *Pretrained Models:* | |
| MLE (Graves, 2013) | 3.87 |
| SeqGAN (Yu et al., 2017b) | 3.84 |
| LeakGAN (Guo et al., 2018) | 3.45 |
| *Non-pretrained Models:* | |
| SeqGAN w/o pretrain | 3.44 |
| LeakGAN w/o pretrain | 7.28 |
| ScratchGAN (de Masson d'Autume et al., 2019) | 3.91 |
| **NAGAN (dropout=0.25)** | 3.54 |
| **NAGAN (dropout=0.2)** | 3.50 |
| **NAGAN (dropout=0.15)** | 3.38 |
| **NAGAN (dropout=0.1)** | **3.23** |

Table 11: Generation performance on COCO and SNLI datasets with mean and variance of three runs with different random seeds.

| Model | COCO | | | | |
|---|---|---|---|---|---|
| | LM Score $\downarrow$ | BLEU$_F$ $\uparrow$ | BLEU$_B$ $\uparrow$ | BLEU$_{HA}$ $\uparrow$ | FED $\downarrow$ |
| **NAGAN (dropout=0.25)** | 3.69$\pm$0.05 | 0.315$\pm$0.007 | 0.257$\pm$0.013 | 0.283$\pm$0.011 | 0.108$\pm$0.005 |
| **NAGAN (dropout=0.20)** | 3.54$\pm$0.05 | 0.342$\pm$0.008 | 0.256$\pm$0.011 | 0.293$\pm$0.010 | 0.111$\pm$0.005 |
| **NAGAN (dropout=0.15)** | 3.41$\pm$0.07 | 0.371$\pm$0.018 | 0.252$\pm$0.014 | 0.301$\pm$0.015 | 0.118$\pm$0.005 |
| **NAGAN (dropout=0.10)** | 3.30$\pm$0.07 | 0.391$\pm$0.023 | 0.245$\pm$0.017 | 0.301$\pm$0.020 | 0.128$\pm$0.011 |
| **NAGAN (dropout=0.00)** | 3.21$\pm$0.11 | 0.415$\pm$0.048 | 0.221$\pm$0.022 | 0.289$\pm$0.030 | 0.191$\pm$0.067 |
| **NAGAN (no training dropout)** | 3.90$\pm$0.11 | 0.267$\pm$0.012 | 0.192$\pm$0.012 | 0.223$\pm$0.011 | 0.160$\pm$0.054 |

| Model | SNLI | | | | |
|---|---|---|---|---|---|
| | LM Score $\downarrow$ | BLEU$_F$ $\uparrow$ | BLEU$_B$ $\uparrow$ | BLEU$_{HA}$ $\uparrow$ | FED $\downarrow$ |
| **NAGAN (dropout=0.25)** | 3.96$\pm$0.15 | 0.279$\pm$0.018 | 0.205$\pm$0.003 | 0.236$\pm$0.004 | 0.059$\pm$0.011 |
| **NAGAN (dropout=0.20)** | 3.78$\pm$0.15 | 0.310$\pm$0.021 | 0.205$\pm$0.004 | 0.246$\pm$0.004 | 0.067$\pm$0.014 |
| **NAGAN (dropout=0.15)** | 3.61$\pm$0.15 | 0.338$\pm$0.019 | 0.203$\pm$0.006 | 0.253$\pm$0.001 | 0.078$\pm$0.019 |
| **NAGAN (dropout=0.10)** | 3.48$\pm$0.13 | 0.363$\pm$0.019 | 0.196$\pm$0.009 | 0.254$\pm$0.003 | 0.096$\pm$0.027 |
| **NAGAN (dropout=0.00)** | 3.48$\pm$0.37 | 0.376$\pm$0.045 | 0.149$\pm$0.032 | 0.213$\pm$0.040 | 0.265$\pm$0.129 |
| **NAGAN (no training dropout)** | 4.44$\pm$0.14 | 0.313$\pm$0.024 | 0.189$\pm$0.004 | 0.235$\pm$0.010 | 0.081$\pm$0.015 |

The hyper-parameters of NAGAN are shown in Table 9, which are tuned to minimize the FED on the validation set. We report the generator architecture, numbers of parameters for the generator and the discriminator in Table 12. The Transformer baseline and NAGAN has the same number of Transformer blocks, attention heads, and size of hidden features, except two differences: the baseline is an autoregressive generator trained by MLE; NAGAN does not need word embeddings because it receives the latent variable $\boldsymbol{Z}$ as input. Unless otherwise specified, the other hyper-parameters of baselines are set according to the official implementation[11]. The reported results in Table 2 are averaged over three runs with different random seeds. As a supplement, we report the result with mean and variance in Table 11. We also show some generated sentences in Table 13.

### A.4.4 IMPLEMENTATION OF ABLATION MODELS

**Autoregressive (AR) Generators**. In Section 4.2, we compare NAGAN against two models with AR generators. These two models differ from NAGAN only in the generator, which is implemented by an autoregressive Transformer with the same architecture except for the attention mask.

AR(Soft) is implemented following Chen et al. (2018). At step $t$, AR(Soft) predicts the next word distribution $\boldsymbol{P}_t = P(\boldsymbol{x}_t | \boldsymbol{x}_{<t}, \boldsymbol{z}) \in \mathbb{R}^V$. Then we obtain the soft word embedding $\boldsymbol{e}_t = \boldsymbol{E}\boldsymbol{P}_t$, where $\boldsymbol{E} \in \mathbb{R}^{E\_dim \times V}$ is the word embedding matrix. After that, $\boldsymbol{e}_t$ is concatenated with the latent variable

---

[11] IRL: https://github.com/FudanNLP/Irl_gen.
SeqGAN, LeakGAN: https://github.com/geek-ai/Texygen.
RelGAN: https://github.com/weilinie/RelGAN.
ScratchGAN: https://github.com/deepmind/deepmind-research/tree/master/scratchgan.
FMGAN: https://github.com/vijini/FM-GAN.

Table 12: Parameter size of all models when vocabulary size is 4838. Note that NAGAN do not need word embeddings because it receives latent variables as inputs.

Table 13: Cases of unconditional generation on the COCO dataset.

| Model | Generator | #Params of Gen | #Params of Dis |
|---|---|---|---|
| GRU | 256-dim GRU | 3.1M | N/A |
| Transformer | AR Transformer | 3.0M | N/A |
| SeqGAN | 128-dim LSTM | 1.4M | 0.9M |
| LeakGAN | Specially Designed | 41.0M | 1.0M |
| IRL | 128-dim LSTM | 2.0M | 5.4M |
| RelGAN | Specially Designed | 4.9M | 3.3M |
| FMGAN | 128-dim LSTM | 1.7M | 3.8M |
| ScratchGAN | 2-layer 512-dim LSTM | 15.4M | 2.3M |
| **NAGAN** | NAR Transformer | 2.0M | 2.2M |

A group of people walking in the grass covered field .
A man riding a motorcycle on a snowy day .
A young man riding a skate board down a hill .
A woman is sitting on a table with a brown laptop .
A group of people standing next to a bunch of bananas .
Two people are sitting on the side of a road .
A big teddy bear laying on a rock in a field .
A group of young man riding a horse in the rain .
A stop sign with a city street .
A plate filled with chicken , and a salad in a basket sitting on a table .

$z \in \mathbb{R}^{Z\_dim}$, and fed into the Transformer as the next input. The latent variable is sampled from the normal distribution $\mathcal{N}(0, I)$. Note that the model generates a sequence of embeddings in training, which does not represent any discrete sentence. We follow Chen et al. (2018) and use greedy decoding to generate texts in inference.

AR(Gumbel) is implemented following Kusner & Hernández-Lobato (2016). AR(Gumbel) predicts the next word distribution at step $t$. Then we obtain the next word in one-hot representation with the Gumbel-softmax tricks (Jang et al., 2017):

$$o_t = \text{onehot}(\arg\max_v (\log P(x_t = v | x_{<t}, z) + g_v)) \in \{0, 1\}^V. \tag{29}$$

The Gumbel-softmax trick is a reparameterized trick, where $o_t$ can be regarded as the one-hot representation of a random sample from the categorical distribution $P(x_t | x_{<t}, z)$ (Jang et al., 2017). The straight-through estimator is also adopted to obtain gradients for $o_t$. After that, $o_t$ is converted to a word embedding $e_t$, which is then similarly concatenated with $z$ and fed into the Transformer as the next input. AR(Gumbel) uses random decoding in inference and thus keeps the same behavior between training and test.

**Other Discriminator Architectures.** In Section 4.2, we equip NAGAN with two different discriminators.

The CNN discriminator is composed of 12 dilated residual blocks (Yu et al., 2017a), where each block sequentially contains a batch normalization layer (Ioffe & Szegedy, 2015), a 1D dilated convolutional layer, and a residual connection (He et al., 2016). The channel size is 300, and the kernel size is 3. The dilation of the first 4 convolutional layers is set to $[1, 1, 2, 3]$, which is repeated two times for the left 8 layers. The convolutional layer does not change the sequence length, where the feature is then fed into a max-pooling layer and an MLP like Eq (10).

The RNN discriminator is a one-layer bidirectional GRU with 256 hidden cells. The encoded feature is also processed by Eq (10) to obtain the final score $\mathcal{D}(O)$.

## A.5 APPLICATION I: MANIPULATE SENTENCES IN LATENT SPACE

In the Offset Vector method (OV) (Zhao et al., 2018), we first randomly generate 100k sentences and obtain the offset vector $\Delta = \overline{Z_t} - \overline{Z_s}$, where $\overline{Z_t}$ is the average of latent variables whose corresponding sentences containing the target word, and $\overline{Z_s}$ is that for the source word. Then, we modify the latent variable by $Z_s := Z_s + \beta \Delta$ until the sentence generated from $Z_s$ contains the target word or the process exceeds a pre-specified iteration number.

Our proposed method, called gradient descent (GD), edits the sentence by applying gradient descent over the latent variable. As described in Section 3.4, we maximize $o_{t, x'}$, where $t$ is the edited position and $x'$ is the target word. In addition to the maximization term, we also add another term to keep the other parts unchanged, which can be formulated as maximizing $\sum_{i \neq t} o_{i, x_i}$, where $x_i$ is the $t$-th token in the source sentence. The final update step is

$$Z_s := Z_s + \frac{\partial(o_{t, x'} + \beta \sum_{i \neq t} o_{i, x_i})}{\partial Z_s}, \tag{30}$$

where $\beta = 0.1$ in our experiments. The update step needs a pre-specified position $t$, where we enumerate all editing positions $t \in [1, L]$ and choose a successful one with the smallest changes. We

also find using different optimizers rather than the simple Gradient Descent optimizer can improve the performance, where we try AdamKingma & Ba (2015) and SGD with different learning rates.

## A.6 APPLICATION II: UNSUPERVISED DECIPHERMENT

### A.6.1 IMPLEMENTATION DETAILS

Unsupervised decipherment is a task similar to text style transfer, where we learn a sequence-to-sequence model without parallel data. Following several previous models on style transfer (Zhu et al., 2017; Yang et al., 2018; Dai et al., 2019), we adapt NAGAN to the task by introducing an encoder. An overview of the model is shown in Figure 6, which can be formulated as:

$$F_{\mathcal{X} \to \mathcal{Y}} := \mathcal{G}(E(\boldsymbol{x}, c_{\mathcal{X}}), c_{\mathcal{Y}}), \quad F_{\mathcal{Y} \to \mathcal{X}} := \mathcal{G}(E(\boldsymbol{y}, c_{\mathcal{Y}}), c_{\mathcal{X}}). \tag{31}$$

$c_{\mathcal{X}}$ and $c_{\mathcal{Y}}$ are labels that specify the input type for the encoder and the desired type for the generator. Note $\mathcal{G}$ is a non-autoregressive generator, where the output length is equal to the source length. (The equal length is a requirement of the task.)

For the reconstruction loss and the cycle loss, we use:

$$\mathcal{L}_{rec,\mathcal{X}} = \mathbb{E}_{\boldsymbol{x} \sim P_{\mathcal{X}}}[d(\widetilde{\boldsymbol{x}}, \mathcal{G}(E(\boldsymbol{x}, c_{\mathcal{X}}), c_{\mathcal{X}}))]; \tag{32}$$
$$\mathcal{L}_{cyc,\mathcal{X}} = \mathbb{E}_{\boldsymbol{x} \sim P_{\mathcal{X}}}[d(\boldsymbol{x}, F_{\mathcal{Y} \to \mathcal{X}}(F_{\mathcal{X} \to \mathcal{Y}}(\boldsymbol{x})))]. \tag{33}$$

$\widetilde{\boldsymbol{x}}$ in Eq (32) is a noisy text modified from $\boldsymbol{x}$, where we randomly drop and swap the order of some tokens (He et al., 2020). $d(\boldsymbol{x}, \mathcal{G}(\cdot))$ is a distance function, where we adopt the cross-entropy:

$$d(\boldsymbol{x}, \mathcal{G}(\cdot)) = -\sum_{t=1}^{L} \log(\text{softmax}(\boldsymbol{s}_t)|_{\boldsymbol{x}_t}), \tag{34}$$

where $\boldsymbol{s}_t$ is defined in Eq (8), and $\text{softmax}(\cdot)|_{\boldsymbol{x}_t}$ means the $\boldsymbol{x}_t$-th dimension of the vector after softmax. The cross-entropy loss is commonly used in previous non-autoregressive generators (Gu et al., 2018).

For the adversarial loss, we utilize a discriminator based on a language model following Yang et al. (2018), which is proven effective on style transfer tasks. The discriminator is trained by a cross-entropy based loss (de Masson d'Autume et al., 2019):

$$\mathcal{L}_{\mathcal{D},\mathcal{X}} = -E_{\boldsymbol{x} \sim \mathcal{X}} \left[ \frac{1}{L_{\boldsymbol{x}}} \sum_{t=1}^{L_{\boldsymbol{x}}} \log P_{\mathcal{D}}(\boldsymbol{x}_t | \boldsymbol{x}_{<t}, c_{\mathcal{X}}) \right] + E_{\boldsymbol{x}' \sim \mathcal{X}'} \left[ \frac{1}{L_{\boldsymbol{x}'}} \sum_{t=1}^{L_{\boldsymbol{x}'}} \log(1 - P_{\mathcal{D}}(\boldsymbol{x}_t' | \boldsymbol{x}_{<t}', c_{\mathcal{X}})) \right]. \tag{35}$$

In Eq (35), $\mathcal{X}'$ is the distribution from the generator, where $\boldsymbol{x}'$ is obtained following two steps: $\boldsymbol{y} \sim P_{\mathcal{Y}}, \boldsymbol{x}' = F_{\mathcal{X} \to \mathcal{Y}}(\boldsymbol{y})$. The adversarial loss for the generator is defined as:

$$\mathcal{L}_{adv,\mathcal{X}} = -E_{\boldsymbol{x}' \sim \mathcal{X}'} \left[ \frac{1}{L_{\boldsymbol{x}'}} \sum_{t=1}^{L_{\boldsymbol{x}'}} \log P_{\mathcal{D}}(\boldsymbol{x}_t' | \boldsymbol{x}_{<t}', c_{\mathcal{X}}) \right]. \tag{36}$$

By swapping the roles of $\mathcal{X}$ and $\mathcal{Y}$, we can obtain the other losses: $\mathcal{L}_{cyc,\mathcal{Y}}, \mathcal{L}_{adv,\mathcal{Y}}, \mathcal{L}_{rec,\mathcal{Y}}$, and $\mathcal{L}_{\mathcal{D},\mathcal{Y}}$. The final losses can be formulated as follows:

$$\mathcal{L}_{\mathcal{G}} = \mathcal{L}_{cyc,\mathcal{X}} + \mathcal{L}_{cyc,\mathcal{Y}} + \alpha \mathcal{L}_{adv,\mathcal{X}} + \alpha \mathcal{L}_{adv,\mathcal{Y}} + \beta \mathcal{L}_{rec,\mathcal{X}} + \beta \mathcal{L}_{rec,\mathcal{Y}}, \tag{37}$$
$$\mathcal{L}_{\mathcal{D}} = \mathcal{L}_{\mathcal{D},\mathcal{X}} + \mathcal{L}_{\mathcal{D},\mathcal{Y}}. \tag{38}$$

where $\alpha, \beta$ are hyper-parameters. $\mathcal{G}$ and $\mathcal{D}$ are optimized alternatively by gradient-based methods.

We utilize a Transformer as the encoder, whose input is the sum of the source sentence embedding and an embedding for the label $c_{\mathcal{X}}$ (or $c_{\mathcal{Y}}$). The output of the Transformer is regarded as the latent variable $\boldsymbol{Z} \in \mathbb{R}^{Z\_dim \times L}$ in Figure 6. The architecture is the same as the Transformer in the discriminator described in Appendix A.3.2.

We adopt a Transformer-based language model as the discriminator, where the input is also added with the label embedding. The architecture of the Transformer is the same as the encoder except for the attention mask. Note that the discriminator should pass the gradient back to the generator, where we also use one-hot representations of sentences and the straight-through estimator. A previous work adopt the same method, so we refer the interested reader to Tu et al. (2020) for more details.

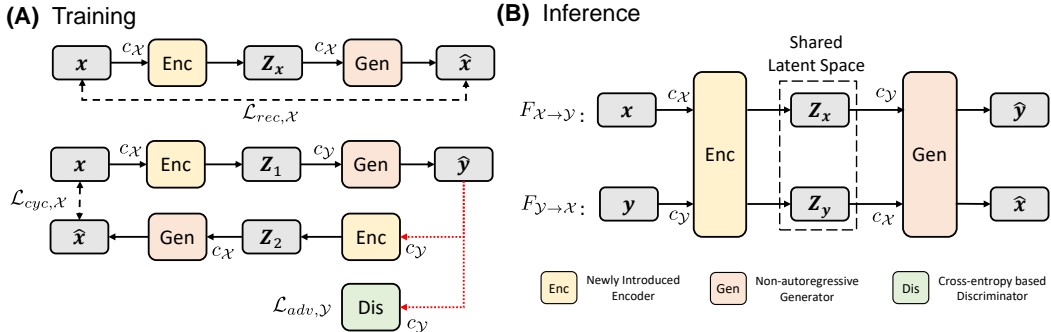

Figure 6: Overview of NAGAN on unsupervised decipherment. (A) Three losses of generators: $\mathcal{L}_{rec,\mathcal{X}}, \mathcal{L}_{cyc,\mathcal{X}}, \mathcal{L}_{adv,\mathcal{Y}}$. The other three losses can be obtained by swapping the roles of $\mathcal{X}$ and $\mathcal{Y}$. Red dotted lines indicate operations bringing the *non-differentiable* problems. (B) The decipher model contains the newly introduced encoder and the non-autoregressive generator, where two types of text are encoded into a shared latent space.

The non-autoregressive generator is kept unchanged, which is the core of our model. Autoregressive generators suffer from the non-differentiable problem when optimizing the cycle loss and the adversarial loss (Red arrows in Figure 6(A)), where our non-autoregressive can be optimized more effectively.

### A.6.2 EXPERIMENT DETAILS

The Word Substitution dataset (Yang et al., 2018) uses the word substitution cipher to encrypt the plaintext. The dataset contains 9,445 words in vocabulary, 200,000 unpaired sentences for both $\mathcal{X}$ and $\mathcal{Y}$ in the training set, and 5,000 sentence pairs in both the validation and the test set. Following previous work (He et al., 2020), we use BLEU-4 for evaluation. We show several cases on the Word Substitution dataset in Table 14.

The BrownW-200 dataset (Gomez et al., 2018) uses the Vigenère cipher for encryption. The dataset contains 200 words in the vocabulary and 51,606 (11,468) sentence pairs for the training (test) set. We break the connections of the paired sentences when training. Following previous work (Chen et al., 2018), we use accuracy for evaluation, which is calculated by the average proportion of words correctly deciphered. The dataset does not contain a validation set, so we choose the best model on the training set as our final model.

For hyper-parameters, we set $\alpha = 1$, $\beta = 1$ at the beginning of the training for the Word Subtituion dataset. Then they decay to 0 linearly in the first 100k batches. (We train the models for 200k batches.) The annealing method is proposed by He et al. (2020). Similarly, we set $\alpha = 0.3$, $\beta = 1$ at the beginning for BrownW-200 dataset. On both datasets, we use the RAdam (Liu et al., 2020) optimizer with a learning rate of 1e-3 to optimize the generator and the discriminator alternatively.

Table 14: Cases of unsupervised decipherment on the WordSub dataset. We show the deciphered text and the golden answer, where the ciphertext is omitted. **Red words** indicates the decipherment errors.

| | |
|---|---|
| **Generated** | **cart** cost is $ num dollars . |
| **Golden** | **entry** cost is $ num dollars . |
| **Generated** | the wait staff and bartenders are rude . |
| **Golden** | the wait staff and bartenders are rude . |
| **Generated** | everyone wants to help . |
| **Golden** | everyone wants to help . |
| **Generated** | all cleaned up we enjoyed the area and **searched** for some drinks and **sunglasses** . |
| **Golden** | all cleaned up we enjoyed the area and **headed** for some drinks and **gaming** . |
| **Generated** | our server **john** was incredible **-** cute , patient , attentive and funny . |
| **Golden** | our server **james** was incredible **:** cute , patient , attentive and funny . |

## A.7 Other Details

For the COCO and SNLI dataset, we train our model for 100 epochs (each epoch contains 1500 batches of samples), evaluate it every epoch, and select the model with the best FED on the validation set. Each training run used approximately 4 Intel Xeon E5-2690 v4 CPUs at 2.60GHz, and 1 Nvidia GeForce RTX 2080 Ti GPU. Although the results of 10-hour training are close to the reported performance (with a gap of 0.01 in terms of $BLEU_{HA}$), we finish the 100 epochs for around 28 hours.

For the synthetic data, we train our model for 50 epochs. Each training run costs around 15 hours with the devices described above.

We also report the latency of one generator training step and the inference in Table 15. Compared with an autoregressive Transformer of the same architecture, NAGAN is slower in training (because our generator's updates need gradients from the discriminator) but faster in inference (6.66x speed up). The fast inference is brought by the parallel decoding. Compared with autoregressive text GANs of the same architecture, NAGAN also has a faster training step because autoregressive GANs need to read its generated tokens as input.

For the unsupervised decipherment task, we train the model for 200k batches, evaluate it every 1.5k batches, and select the model with the best BLEU-4 on the validation set for the Word Subtitution dataset (the best accuracy on the training set for the BrownW-200 dataset). Each training run cost around 37 hours with the same devices described above.

Table 15: Training and inference latency of an autoregressive Transformer and NAGAN on the COCO dataset. All results are evaluated with batch size of 32.

| Model | Generator Training Step | Inference Latency |
|---|---|---|
| MLE-Transformer | 145ms (1.00x) | 213ms (1.00x) |
| NAGAN | 210ms (0.69x) | 32ms (6.66x) |

