# OpenReview forum: "A Text GAN for Language Generation with Non-Autoregressive Generator"
_ICLR.cc/2021/Conference — Reject_

### Official Review · AnonReviewer1 · 2020-10-25
**Good extension of text GAN but limited significance for text generation**

**Rating:** 6
**Confidence:** 4

**Review:**

The paper creatively extends text GAN by introducing non-autoregressive generator, which is a well-known notion in translation and VAE like generation but not often applied in a GAN setting. The paper argues that a non-autoregressive generator brings more effective gradient-based optimization and also good latent representation learning capability.

The comparison between NAGAN and other text GANS reads okay, but the reviewer concerns the limited scope and significance of this paper.

1, Given very strong text generation capability of MLE learning and pre-training, NAGAN makes little contribution to push the generation SOTA. Audiences of this approach are also limited. In this paper, given a very old baseline of MLE and a bunch of text GANs, the overall performance of NAGAN is still not much leading. Let alone compare it to other strong pre-trained generators.

2. When claiming good latent representation learning capability, there should be a big gap between NAGAN and text VAEs in this aspect. If the author adds more control and manipulation experiments in text VAE, NAGAN will be not as shining as now.

3. Non-autoregressive generator has difficulties in generalizing to long text generation and conditional generation. How does the author consider such settings, instead of simple unconditional generation in toy datasets like COCO?

Overall, the reviewer thinks this is a well-written paper, but a boardline one considering its limited significance for the venue.

---

> ### Author Response · Authors · 2020-11-17
> **Author Response Part #2**
>
> ### Question #1
>
> We agree that NAGAN does not push the general text generation SOTA, and it is still far behind the large-scale pretrained models like GPT3. However, NAGAN tries to solve some challenging problems, and we have summarized NAGAN's contributions on three different research directions in Part 1 of our response.
>
> We also mention that MLE still has weaknesses though it is very powerful. The exposure bias problem may lead to repetitions [6]. It is not convenient in some unsupervised settings where we do not have paired text data, such as unsupervised text style transfer. Adversarial training can address the two issues and more, so we believe that text GANs need further studied.
>
> [6] The Curious Case of Neural Text Degeneration. ICLR2020
>
> ### Question #2
>
> **First, you mentioned that there should be a big gap between NAGAN and VAEs' representations, but we do not find evidence that supports the statement.** In our Application I, FMGAN is finetuned after the VAE pretraining. Comparing FMGAN and NAGAN, we can see similar performance on this task if we use the same method (Offset Vector). Moreover, **we add some experiments for exploring the latent space in Appendix A.2**, where we observe that NAGAN and VAEs behave similarly in sentence interpolation.
>
> **Second, NAGAN can be extended to some application that VAE cannot do.** We have summarized our views from the perspective of generation applications: 1. NAGAN supports back-propagation through discrete text data. 2. NAGAN can match two distributions in text space directly. More is discussed in Part 1 of our response.
>
> **Third, we think VAEs are not in conflict with GANs.** We can find many interesting models in image generations that combine GANs and VAEs [8, 9]. These combined models have advantages from both methods and are convenient to be used in more applications such as image editing.
>
> [8] Autoencoding beyond pixels using a learned similarity metric. ICML2016
>
> [9] Neural Photo Editing with Introspective Adversarial Networks. ICLR2017
>
>
>
>
> ### Question #3
>
> We admit that it will be harder to apply non-autoregressive generators in long text generation or conditional generation with a complex dataset. However, we have already shown NAGAN's abilities on some challenging datasets.
>
> In addition to the small COCO dataset, we use the SNLI dataset, which contains 714,667 samples in the training set, and 42,981 words in the vocabulary. It is very challenging for GANs training because the search space increases exponentially according to the vocabulary size. Moreover, in Figure 3, we show that our model is not very sensitive to the sentence length (length varies from 10 to 25), where autoregressive text GANs become worse rapidly as the length increases.
>
> We also test NAGAN on the unsupervised word decipherment task, which can be regarded as an unpaired sequence-to-sequence problem. **NAGAN outperforms all SOTA models which use autoregressive generators.** We show some cases below (more in Table 14), where we find the generated sentences have good fluency, even though some words are not correctly deciphered.
>
> |           |                                                                                      |
> |-----------|--------------------------------------------------------------------------------------|
> | Generated | all cleaned up we enjoyed the area and **searched** for some drinks and **sunglasses** . |
> | Golden    | all cleaned up we enjoyed the area and headed for some drinks and gaming .           |
> | Generated | our server **john** was incredible **-** cute, patient, attentive and funny .            |
> | Golden    | our server james was incredible : cute, patient, attentive and funny .               |
>
>
>
> **Thanks for your attention and please let us know if we can address your concerns.**

---

> ### Author Response · Authors · 2020-11-17
> **Author Response Part #1**
>
> Thanks for your comments. We agree that NAGAN does not push the general generation SOTA. However, we take a step forward in several research directions and provides new insights for some fundamental but challenging problems. We summarize them as follows:
>
> ### From the perspective of text GANs
>
> We provide a new insight that **connects the notorious optimization problem with the implicit generative model and the non-autoregressive generator**. We also successfully equip text GANs with latent variables.
>
> Existing text GANs suffer from the non-differentiable problem, where RL and gradient-based methods are ineffective with or without MLE pretraining. Many works [1, 2] are devoted to the problem by improving RL methods or approximation methods. **To the best of our knowledge, the connection between the non-differentiable problem and the non-autoregressive generator is still untouched in the literature.** We for the first time make the text GANs training from scratch effectively, even though the straight-through estimator does not work well in autoregressive generators.
>
> Moreover, previous text GANs seldomly use latent variables, and directly injecting latent variables into the explicit generative model can be problematic (explained in Section 2.1 and Appendix A.1.1). Therefore, we explore the feasibility of adopting an implicit generative model with a non-autoregressive generator.  The non-autoregressive generator can be applied to various applications conveniently.
>
> ### From the perspective of non-autoregressive text generation
>
> We introduce adversarial training objectives and explore the open-ended generation task.
>
> Existing non-autoregressive models suffer from the multi-modality problem [3] and hard to fit a complex corpus if there are no strong connections between inputs and outputs. Most of them require knowledge distillation to reduce the dataset complexity, otherwise they will experience a serious performance drop [4, 5]. **To our knowledge, no non-autoregressive models have been used in an open-ended generation task**, such as the unconditional generation. Therefore, our work can be regarded as an early exploration of generating diverse sentences with a non-autoregressive generator.
>
> ### From the perspective of generation applications
>
> Our method is powerful and convenient in various tasks, where **we outperform existing SOTA in the unsupervised word decipherment task**.
>
> In Application I, we try to manipulate sentences by back-propagating the gradients through the discrete text. Compared with the OV method, which estimates the offset vector by massive sampling, our idea is more straightforward and achieves a higher success rate.
>
> In Application II, we apply NAGAN to a benchmark of the unsupervised decipherment. Our model outperforms all SOTA models for two reasons: First, text GANs can naturally match two distributions directly in text space, which is hard for VAEs. Second, we tackle the non-differentiable problem that gradients of cycle loss can flow through the generator, which leads to a better alignment.
>
> Our method can also be applied to other applications that need sentence-level optimization, which is convenient in controlled text generation, for example, generating sentences with desired attributes.
>
>
>
>
> In conclusion, although NAGAN does not push the general text generation SOTA, **we believe it contributes to several research directions, provides strong results, novel ideas, and space for further investigation**.
>
> [1] Long Text Generation via Adversarial Training with Leaked Information. AAAI18
>
> [2] Training Language GANs from Scratch. Neurips2019
>
> [3] Non-Autoregressive Neural Machine Translation. ICLR2018
>
> [4] FlowSeq: Non-Autoregressive Conditional Sequence Generation with Generative Flow. EMNLP2019
>
> [5] Understanding Knowledge Distillation in Non-autoregressive Machine Translation. ICLR2020

---

### Official Review · AnonReviewer4 · 2020-10-28
**I think that this paper is above the average because it provides comprehensive experiments but lacks analysis about the generation process**

**Rating:** 6
**Confidence:** 4

**Review:**

This paper introduces the non-autoregressive generator in the GAN-based text generation, making textGAN can be trained without pre-training and better utilize latent variables to control the style of generated text.

Introducing non-autoregressive architectures into GAN-based text generator is a natural idea, and the modelling ability of non-autoregressive generator has been verified at BERT. I think that this paper is above the average because it provides comprehensive experiments (including unconditional text generation, unsupervised decipherment and sentence manipulation), showing the significant improvement in various evaluation metrics compared to the text model without NAR and baselines.

However, this paper should have more analysis of how non-autoregressive architectures work in the GAN-based text generation.
1.	How latent variables in the non-autoregressive generator influence the content of the generated text? Can you provide some analysis or examples about it? (e.g. change the value of some latent variable continually (from 0 to 1?) and give the generated text).
2.	Can you give some analysis about the generation process of the non-autoregressive generator, (e.g. attention map), which makes the generator more interpretable.
3.	The user study is absent.
4.	Is dropout necessary for the non-autoregressive generator? What if the dropout rate is 0, how the performance of generator changes?
5.    Can you give more details about experiments, such as model parameters, training time, inference time and GPU you used?

---

> ### Author Response · Authors · 2020-11-17
> **Authore Response Part #2**
>
> ### Question #4
>
> We add two experiments: we set dropout rate to 0 in inference (the dropout rate in training is kept unchanged as 0.25); we remove dropout in training. The results are added to Table 2 and Table 11. Some results are shown here:
>
>
> | MODEL on COCO                 | LM Score | BLEU_F | BLEU_B | BLEU_HA | FED   |
> |-------------------------------|----------|--------|--------|---------|-------|
> | NAGAN (dropout=0.25)          | 3.69     | 0.315  | 0.257  | 0.283   | 0.108 |
> | NAGAN (dropout=0.20)          | 3.54     | 0.342  | 0.256  | 0.293   | 0.111 |
> | NAGAN (dropout=0.10)          | 3.30     | 0.391  | 0.245  | 0.301   | 0.128 |
> | NAGAN (dropout=0.00)          | 3.21     | 0.415  | 0.221  | 0.289   | 0.191 |
> | NAGAN (no dropout in training)| 3.90     | 0.267  | 0.192  | 0.223   | 0.160 |
>
> For NAGAN (dropout=0), it agrees with our conclusion in Section 4.1 that the fluency continues to increase, and the diversity declines when the dropout rate becomes smaller.
>
> For NAGAN (no dropout in training), we observe spiky losses and unstable training process, but it still outperforms some baselines without MLE pretraining. The phenomenon has been found in image GANs [7, 8], but not significant in autoregressive baselines.
>
> As an explanation, the dropout can be regarded as a method of introducing extra latent variables in GANs training. If you are interested, please refer to Appendix A.1.3 where more discussions are added.
>
> [2] Large Scale GAN Training for High Fidelity Natural Image Synthesis. ICLR2019
>
> [3] A 9k-star repository on Github. https://github.com/soumith/ganhacks.
>
> ### Question #5
>
> We add Table 12 to show the parameter size of all models. We use a very small transformer (128-dim hidden, 5 layers, 4 attention heads), where NAGAN's generator (2.0M parameters) is smaller than the median size of baselines' generator (3.1M parameters).
>
> We add some training details in Appendix A.7 (where we also fix some errors in the previous version). Some details are shown below.
>
> For the COCO and SNLI dataset, we train our model for 100 epochs (each epoch contains 1500 batches of samples). Each training run used approximately 4 Intel Xeon E5-2690 v4 CPUs at 2.60GHz, and 1 Nvidia GeForce RTX 2080 Ti GPU. Although the results of 10-hour training are close to the reported performance (with a gap of $0.01$ in terms of $BLEU_{HA}$), we finish the 100 epochs for around 28 hours.
>
> We also report the latency of one generator training step and the inference in the table below. Compared with an autoregressive Transformer of the same architecture, NAGAN is slower in training (because our generator's updates need gradients from the discriminator) but faster in inference (6.66x speed up). The fast inference is brought by the parallel decoding.
>
>
> | Model           | Generator Training Step | Inference Latency |
> |-----------------|-------------------------|-------------------|
> | MLE-Transformer | 145ms (1.00x)           | 213ms (1.00x)     |
> | NAGAN           | 210ms (0.69x)           | 32ms (6.66x)      |
>
>
>
> **Please feel free to let us know if you still have some concerns or questions**.

---

> ### Author Response · Authors · 2020-11-17
> **Author Response Part #1**
>
> Thanks for your comments. We add some experiments to explore how the non-autoregressive generator works. Please find our responses below and **feel free to let us know if you still have some concerns or questions**.
>
> ### Question #1
>
> We add three experiments to explore the properties of latent space in Appendix A.2.1 ~ A.2.3. Please see the revised paper for images and a better format.
>
> * In Appendix A.2.1, we show several examples of sentence interpolation where the sentences are generated from a group of linearly interpolated latent variables.
> * In Appendix A.2.2, we try to concatenate a newly sampled latent variable to an existing latent sequence and translate the sequences to the generated sentences.
> * In Appendix A.2.3, we investigate whether the latent variable at position $i$ is highly related to the token at the same position. However, we find latent variables at different positions plays a similar role in generating the $i$-th token, regardless of their position. This property is not desired and can make NAGAN less interpretable. However, we think that adding regularizers will be a good way to make the tokens more correlated to nearby latent variables, which is left for future work.
>
> ### Question #2
>
> In Appendix A.2.4, we add an experiment to show how the non-autoregressive generator generates the sequence. Please see the revised paper for images and detailed explainations. From the cosine similarities between intermediate hidden states and the last output, we find two interesting properties:
>
> * The non-autoregressive generator will gradually determine the hidden states as the data flows through the transformer layers.
> * The non-autoregressive generator tends to determine some of the tokens first and then the others. The first determined tokens include definite or indefinite articles at the front, prepositions in the middle, and full stops at the end. This strategy may make the generation of the remained tokens easier, because it is similar to a masked language model when the near context is given.
>
> ### Question #3
>
> We conduct human evaluation following a previous work [1]. Each model generates 100 sentences, and 3 workers on AMT are hired to judge whether the sentence is generated by a machine or written by a human. We show the preliminary results below.
>
> | MODEL on COCO        | score |
> |----------------------|-------|
> | GRU-MLE              | 0.32  |
> | SeqGAN               | 0.21  |
> | NAGAN (dropout=0.25) | 0.44  |
> | Human                | 0.86  |
>
> We can see NAGAN outperforms GRU-MLE and SeqGAN. However, it conflicts with the common conclusion that MLE is a strong baseline which has good diversity and fits the data distribution well.
>
> Unconditional text generation requires the model to capture the corpus distribution, where diversity is very important in this task. Manual evaluation seems not suitable for evaluating diversity. For example, a model can always generate the same sentence for better overall score. We also try to show 10 samples to a worker and ask whether the sentences are diverse. However, manual evaluation still cannot identify the notorious mode collapse problem.
>
> [1] Toward Diverse Text Generation with Inverse Reinforcement Learning. ICJAI2018

---

> ### Author Response · Authors · 2020-11-20
> **Looking forward to your reply !**
>
> We are looking forward to your reply and very willing to discuss if you still have concerns.
> We kindly remind that the author discussion period will end soon, and we are not able to post comments after **Nov. 24 (Tuesday)**.

---

### Official Review · AnonReviewer3 · 2020-10-29
**Interesting work on integrating non-autoregressive generator to text GAN**

**Rating:** 6
**Confidence:** 4

**Review:**

**Summary**
This paper proposed a new text GAN framework by combining non-autoregressive text generator based on transformer, straight-through gradient approximation, and various regularization techniques such as gradient penalty and dropout. The paper demonstrates the superiority of non-autoregressive generator in the context of text GANs through various experiments including unconditional text generation, latent space manipulation and unsupervised decipherment.

**Pros**
- This work narrows the gap between image GAN and text GAN by leveraging recent advances on non-autoregressive text generators and a straight-through gradient approximation. While these components are well studied in previous works, I think this work presents a neat combination of them in order to solve a well-known problem.

- The paper provides rich discussions in training text GAN and comprehensive experiments and ablations to demonstrate the usefulness of an implicit text generator in different contexts.

**Concerns & Questions to Answer during rebuttal**
- The original text GAN papers were mainly motivated to address the *exposure bias* problem in maximum likelihood estimation for autoregressive generators. In other words, when we use an autoregressive generator to sequentially generate tokens one by one, there is a distribution mismatch between training and test phase. In your case, now that you already have a non-autoregressive text generator, is there any *theoretical motivation/insights* for using the adversarial training framework? We know that MLE is statistically efficient (achieving Cram´er–Rao Lower Bound) and possesses many good properties, maybe training the non-autoregressive text generator with MLE (e.g., FlowSeq [1] or variational inference) is a better choice?

- The non-autoregressive (NA) text generator have been well studied recently so the novelty of this work is more on the integration of NA generator with adversarial training. Thus the main challenge here is how to solve the non-differentiability problem. The paper directly leverages a traditional workaround, the straight through estimator, which is a biased gradient approximation. Is the bias going to be an issue and is there any better strategy? I think the paper need to provide more discussions on this aspect. Overall the method section need to be polished with more details, as I feel this part is currently hard to follow.

- In figure 1, $z_1, \ldots, z_L$ are sampled independently, which are sent to a transformer and later produced the sample. Is the independence between $z_1, \ldots, z_L$ going to be a problem? When we use transformer to do neural machine translation, the attention mechanism will capture the dependence in the input sentence ($z_1, \ldots, z_L$ in this context) and then produce the output correspondingly. Hence will the independence in $z_1, \ldots, z_L$ lead to a less expressive sample distribution in your text generator (although this is not an issue in image GAN)?

- The paper also propose to use the Max Gradient Penalty from image GAN domain. The Max Gradient Penalty was introduced under the framework of Wasserstein GAN or Lipschitz GAN framework, which aims to constraint the function space to be Lipschitz smooth. However this work uses the vanilla GAN objective (eq 12 and eq 13), which is not the WGAN or LGAN framework. Thus the regularization may not be theoretically correct. Also why not instead use the WGAN objective which is empirically more stable and theoretically sound?

- Experiments: In table 2, all the results for NAGAN use the dropout with a positive ratio. How does NAGAN perform without dropout? Also I wonder if the comparison in the table and figures are fair, since most previous methods.baselines such as MLE or SeqGAN only use a vanilla RNN/LSTM, while NAGAN has a more complicated structure with transformers, and additional regularization such as gradient penalty and dropout. Perhaps we should control at least the number of parameters to be in the same level.

[1] FlowSeq: Non-Autoregressive Conditional Sequence Generation with Generative Flow

Update after rebuttal:
After seeing the author response below, no change to my score.

---

> ### Author Response · Authors · 2020-11-17
> **Author Response Part #2**
>
> ## Question #4
>
> The vanilla GAN objective is a special case of Lipschitz GAN [6], where $\phi(x)=\varphi(-x)=\psi(-x)=-\log(\sigma(-x))$ in their Equation (10). The formulation is mentioned just before Section 4.1 in their paper. Their experiments also show that vanilla objectives with the MaxGP penalty perform similarly with other objectives, including the WGAN objective.
>
> [6] Lipschitz Generative Adversarial Nets. ICML2019
>
> ## Question #5
>
> ### The problem of the dropout.
>
> We add two experiments: we set the dropout rate to 0 in inference (the dropout rate in training is kept unchanged as 0.25); we remove the dropout in training. The results are added to Table 2 and Table 11. Some results are shown here:
>
> | MODEL on COCO                 | LM Score | BLEU_F | BLEU_B | BLEU_HA | FED   |
> |-------------------------------|----------|--------|--------|---------|-------|
> | NAGAN (dropout=0.25)          | 3.69     | 0.315  | 0.257  | 0.283   | 0.108 |
> | NAGAN (dropout=0.20)          | 3.54     | 0.342  | 0.256  | 0.293   | 0.111 |
> | NAGAN (dropout=0.10)          | 3.30     | 0.391  | 0.245  | 0.301   | 0.128 |
> | NAGAN (dropout=0.00)          | 3.21     | 0.415  | 0.221  | 0.289   | 0.191 |
> | NAGAN (no dropout in training)| 3.90     | 0.267  | 0.192  | 0.223   | 0.160 |
>
> For NAGAN (dropout=0), the result agrees with our conclusion in Section 4.1 that the fluency continues to increase, and the diversity declines when the dropout rate becomes smaller.
>
> For NAGAN (no dropout in training), we observe spiky losses and unstable training process, but it still outperforms some baselines without MLE pretraining. The phenomenon has been found in image GANs [7, 8], but not significant in autoregressive baselines.
>
> As an explanation, the dropout can be regarded as a method of introducing extra latent variables in GANs training. If you are interested, please refer to Appendix A.1.3 where more discussions are added.
>
> [7] Large Scale GAN Training for High Fidelity Natural Image Synthesis. ICLR2019
>
> [8] A 9k-star repository on Github. https://github.com/soumith/ganhacks.
>
>
>
>
> ### The problem of the parameter size.
>
> We use a very small transformer (128-dim hidden, 5 layers, 4 attention heads), and NAGAN does not have much more parameters than baselines. We present the numbers of parameters in Table 12, where **NAGAN's generator (2.0M parameters) is smaller than the median size of baselines' generator (median: 3.1M parameters)**. (ScratchGAN uses 2-layer LSTM with 512 cells, where we made a wrong statement previously. It has been fixed in Appendix A.4.3.)
>
> We do not use an unified architecture for our baselines because the GAN training is very sensitive to the generator's architecture and hyper-parameters.  A research confirms our observation and they find that the transformer is very unstable in RL training [9].
>
> In Table 2, we add an MLE-trained autoregressive transformer as our baseline, which also has 128-dim hidden states, 5 layers, and 4 attention heads. We also try a SeqGAN with the same generator, but it does not outperform the one with GRU. The results are shown below, and our conclusion does not change.
>
> | MODEL on COCO      | LM Score | BLEU_F | BLEU_B | BLEU_HA | FED   |
> |--------------------|----------|--------|--------|---------|-------|
> | GRU                | 4.13     | 0.292  | 0.325  | 0.308   | 0.094 |
> | Transformer        | 3.90     | 0.327  | 0.321  | 0.324   | 0.094 |
> | SeqGAN LSTM        | 4.03     | 0.298  | 0.285  | 0.291   | 0.108 |
> | SeqGAN Transformer | 4.47     | 0.246  | 0.253  | 0.250   | 0.114 |
>
> [9] Stabilizing Transformers for Reinforcement Learning. Arxiv2019
>
>
>
> ### The problem of the gradient penalty.
>
> The gradient penalty is designed for gradient-based optimization, where many baselines are using RL. There is no theoretical motivation to adopt the gradient penalty on them.
>
> RelGAN and FMGAN are optimized by gradient-based method, where RelGAN uses the Gumbel-Softmax approximation, and FMGAN uses the Soft-Embedding approximation. In Section 4.2, we conduct ablation studies using an autoregressive generator with the two approximation methods, where the gradient penalty are adopted in all models. As shown in Figure 3 and Table 3, NAGAN remarkably outperform these two approximation methods.
>
>
>
> **Please feel free to let us know if you still have some concerns or questions**.

---

> ### Author Response · Authors · 2020-11-17
> **Author Response Part #1**
>
> Thank you for your constructive comments, which point out some important intuitions behind the models and give us a chance to clarify them. Please find our responses below and **feel free to let us know if you still have some concerns or questions**.
>
> ### Question #1
>
> Existing non-autoregressive models suffer from the multi-modality problem [1] and are hard to fit a complex corpus if there are no strong connections between inputs and outputs. Most of them, including FlowSeq, require knowledge distillation to reduce the dataset complexity, otherwise they will experience a serious performance drop [2, 3]. Theoretically, introducing latent variables can alleviate the multi-modality problem, but it seems not trivial empirically from existing evidence.
>
> We believe that a key problem is the independence assumption of MLE objectives, which forces the model to generate each target token on a specific position regardless of tokens on the other positions. It can lead to over-corrections [4] and word repetitions. However, GANs objectives work as a sentence-level loss, where the discriminator will consider the whole sequence and punish implausible sentences, e.g., sentences with repeated words.
>
> Moreover, GANs can apply to some applications that cannot be easily achieved by a VAE. In unsupervised text decipherment, we show that GANs can directly align two distributions (the deciphered text and the plaintext) in the text space. However, it is difficult to apply VAEs in this task, where we may still face the non-differentiable problem.
>
> We update Section 2.3 with the discussion above.
>
> [1] Non-Autoregressive Neural Machine Translation. ICLR2018
>
> [2] FlowSeq: Non-Autoregressive Conditional Sequence Generation with Generative Flow. EMNLP2019
>
> [3] Understanding Knowledge Distillation in Non-autoregressive Machine Translation. ICLR2020
>
> [4] Minimizing the Bag-of-Ngrams Difference for Non-Autoregressive Neural Machine Translation. AAAI2020
>
> ### Question #2
>
> Yes, it is exactly a challenge to tackle the non-differentiable problem, where a good estimator is important. **However, in our paper, instead of focusing on the estimator, we find that the generator architecture has a key role in alleviating the non-differentiable problem in text GANs training.**
>
> We have shown that adopting a non-autoregressive generator as the implicit generative model can bring effective gradient-based training, even if the estimator is biased and does not work well in the autoregressive generator. **To the best of our knowledge, our work is the first to reveal the connection between the non-differentiable problem and the non-autoregressive generator.**
>
> We add a simple explanation at the end of Section 2.2. To obtain gradients, an autoregressive generator with the gradient-based optimization method has to apply the gradient estimator recurrently, because the generator reads discrete tokens as inputs. In other words, the gradient flow from the last token will be approximated multiple times when it reaches the start of the sequence. On the contrary, a non-autoregressive generator has a feed-forward structure, and only one approximation is required at the last layer of the generator. As demonstrated in Section 4.2, NAGAN outperforms autoregressive models in gradient-based optimization, and the autoregressive models become worse rapidly when the sentence length grows.
>
> We refine our method parts and hope that we have made it clear.
>
> [5] Language GANs Falling Short. ICLR2020
>
> ### Question #3
>
> Yes, we agree that a complex prior distribution can enhance the expressive ability of the model, but we just choose the simplest independent normal distribution, which is also adopted by FlowSeq [2]. **Moreover, the simplest distribution reveals a new insight**: the non-autoregressive generator can be trained to generate diverse text, even if the inputs and the generated sentences are not strongly related. Most works on non-autoregressive models are focusing on machine translation and suffer from the multi-modality problem, and no experiments are conducted on an open-ended task like the unconditional generation.
> Therefore, our work can be regarded as an early exploration of generating diverse sentences with a non-autoregressive generator.
>
> Back to the choice of the prior distribution, we agree that it is important to investigate a more complex prior distribution, which is left as our future work.
>
> [2] FlowSeq: Non-Autoregressive Conditional Sequence Generation with Generative Flow. EMNLP2019

---

> ### Author Response · Authors · 2020-11-20
> **Looking forward to your reply !**
>
> We are looking forward to your reply and very willing to discuss if you still have concerns.
> We kindly remind that the author discussion period will end soon, and we are not able to post comments after **Nov. 24 (Tuesday)**.

---

### Author Response · Authors · 2020-11-17
**We uploaded the first revision of our paper**

We have uploaded a revision of our paper, and the modified part is highlighted in yellow.

**We summarize the modifications here:**

* At the end of Section 2.2, we explain **why the non-autoregressive generator can benefit gradient-based optimization**.
  * TL;DR. Autoregressive models need to apply the gradient estimator recurrectly, but our model applies the gradient estimator only at the output of the generator.
* In Section 2.3, we discuss why we introduce the GANs objective to non-autoregressive generation.
  * TL;DR. GANs objective does not rely on the independence assumption, which may alleviate the multi-modality problem.
* In Section 3, we refine the descriptions of our methods.
* In Table 2, we add an MLE-trained Transformer as our baseline. We also add **the result of NAGAN (dropout=0)**. It agrees with our previous conclusion.
* In Appendix A.1.3, we add discussions about dropouts. We answer two questions: **why the dropout can stabilize training, why the dropout can balance fluency and diversity**.
  * Dropout brings diversity, which avoids mode collapse.
  * Dropout can be regarded as a method of introducing extra latent variables. A small dropout rate means a concentrated distribution with smaller variance.
* In Appendix A.2, we add four case studies about the latent space and the generation process.
  * We show several examples of sentence interpolation where the sentences are generated from a group of linearly interpolated latent variables.
  * We try to concatenate a newly sampled latent variable to an existing latent sequence and translate the sequences to the generated sentences.
  * We investigate whether the latent variable at position $i$ is highly related to the token at the same position. The answer is no.
  * We investigate the generation process. From the cosine similarities between hidden states, we find the model gradually determine the hidden states as the data flows through the transformer layers.
* In Table 12 and Appendix A.4.3, we add the parameter size of all models. (ScratchGAN uses a 2-layer LSTM with 512 cells, where we made a wrong statement previously)
  * NAGAN's generator (2.0M parameters) is small than the median size of baselines (3.1M parameters).

**Some minor revisions:**

* In Table 11, we add the results of NAGAN(dropout=0) and NAGAN(no training dropout) with mean and variance.
* In Appendix A.7, we add the model latency and fix some wrong data about the training time.

---

### Decision · Program_Chairs · 2021-01-07
**Final Decision**

**Decision:**

Reject

**Comment:**

This paper proposes GAN-training of a non-autoregressive generator for text. To circumvent the usual problems with non-differentiability of text GANs, the authors turn to Gumbel-Softmax parameterisation and straight-through estimation.

There are a number of aspects to this submission and they are not always clearly positioned. I will concentrate on the two aspects that seem most crucial:

1. The authors position their generator as an implicit generator, but it really isn't. If we take the continuous interpretation of the output distributions: the Gumbel-Softmax transformation does correspond to a tractable density, the Concrete density of Maddison et al, with known parameter. If we take the discrete interpretation of the output distribution: Gumbel-argmax is just an alternative to sampling from a Categorical distribution with known parameter. In either case, the generator maps the noise source to a collection of conditionally independent distributions each of which has a known parameter and analytical density/mass function. The authors do, however, train the architecture using a GAN-type objective *as if* the generator were implicit.

2. In the discussion phase the authors added that GAN training overcomes the independence assumptions made by the generator. Whereas that makes intuitive sense, it suddenly changes the emphasis of the contributions, from proposing an implicit generator (presumably powerful for it being implicit) to proposing a way to circumvent the strong independence assumptions of the generator with a mechanism other than more traditional approximate marginalisation of VAEs. In their rebuttal, the authors commented on the use of non-autoregressive VAEs in neural machine translation, and though those observations have indeed been made, they might well be specific to MT. The simplest and more satisfactory response would be to ablate the use of the GAN objective (that is, to train a non-autoregressive VAE, also note that, with the same choice of likelihood, posterior collapse is rather unlikely to happen).

Other problems raised by reviewers were addressed in the rebuttal, and I would like to thank the authors for that. For example, ablating the non-autoregressive generator and comparing to REINFORCE. I believe these improved the submission.

Still, I cannot recommend this version for publication. I would suggest that the authors consider careful ablations of the components they see as precisely important for the results (that currently seems to be the GAN-like objective despite the model not, strictly speaking, requiring it).